**EMBO** *reports*

# The Dectin-1 and Dectin-2 clusters: C-type lectin receptors with fundamental roles in immunity

Mariano Malamud ✉ & Gordon D Brown ✉

## Abstract

**The ability of myeloid cells to recognize and differentiate endogenous or exogenous ligands rely on the presence of different transmembrane protein receptors. C-type lectin receptors (CLRs), defined by the presence of a conserved structural motif called C-type lectin-like domain (CTLD), are a crucial family of receptors involved in this process, being able to recognize a diverse range of ligands from glycans to proteins or lipids and capable of initiating an immune response. The Dectin-1 and Dectin-2 clusters involve two groups of CLRs, with genes genomically linked within the natural killer cluster of genes in both humans and mice, and all characterized by the presence of a single extracellular CTLD. Fundamental immune cell functions such as antimicrobial effector mechanisms as well as internalization and presentation of antigens are induced and/or regulated through activatory, or inhibitory signalling pathways triggered by these receptors after ligand binding. In this review, we will discuss the most recent concepts regarding expression, ligands, signaling pathways and functions of each member of the Dectin clusters of CLRs, highlighting the importance and diversity of their functions.**

**Keywords** C-type Lectin Receptors; Dectins; Immune Response; Signaling Pathways
**Subject Categories** Immunology; Microbiology, Virology & Host Pathogen Interaction; Signal Transduction

## Introduction

The C-type lectins are a superfamily of proteins originally named by their ability to recognize carbohydrate structures in a $Ca^{2+}$-dependent manner (Drickamer, 1999). These proteins, classified into seventeen groups based on their phylogeny and structural organization, are either membrane-bound or secreted and are characterized by the presence of one or more C-type lectin-like domains (CTLDs), a structure motif formed by two protein loops stabilized by two conserved disulfide bridges at the base of each loop (Zelensky and Gready, 2005). The interaction between $Ca^{2+}$ and conserved amino acids motifs in the CTLD allows carbohydrate binding and determine binding specificity: the EPN (Glu-Pro-Asn) motif permit the binding of mannose-type

carbohydrates, whereas the QPD (Gln–Pro–Asp) motif confers specificity for galactose-type carbohydrate (Zelensky and Gready, 2005). However, many C-type lectins do not contain the elements required for $Ca^{2+}$ binding and can also recognize a diverse range of ligands, including proteins or lipids (Brown et al, 2018).

Transmembrane C-type lectin receptors (CLRs) are one group of Pattern Recognition Receptors expressed by immune cells that contribute to the recognition of both microbial components, known as pathogen-associated molecular patterns (PAMPs), and endogenous host-derived molecules, known as damage-associated molecular patterns (DAMPs) (Gong et al, 2020a). Upon recognition of their ligands, these CLRs can trigger various intracellular signaling pathways that broadly result in the activation or inhibition of cellular function, modulating innate and adaptive immune responses (Fig. 1) (Brown et al, 2018). Based on their intracellular signaling domains, CLRs can be classified into (i) CLRs with immunoreceptor tyrosine-based activation motif (ITAM) domains, (ii) CLRs with hemITAM domains, (iii) CLRs with immunoreceptor tyrosine-based inhibition motif (ITIM) domains or (iv) CLRs without clear ITAM or ITIM domains (Fig. 2) (Sancho and Reis e Sousa, 2012a).

## The Dectin-1 and Dectin-2 clusters

Of particular interest in this review are the groups of receptors encoded in two different genomic regions but both within the natural killer gene complex (NKC) on the human chromosome 12 and the corresponding region in the mouse chromosome 6, termed Dectin-1 and Dectin-2 clusters (Plato et al, 2013). The Dectin-1 cluster is located in the centromeric part of the NKC, while the Dectin-2 cluster is encoded at the telomeric end of the NKC (Fig. 3) (Dambuza and Brown, 2015).

All the receptors included in these clusters are type II transmembrane receptors, where the C-terminal encodes the extracellular region of the protein consisting of a single CTLD and a stalk domain, connected to an intracellular domain via a transmembrane region (Fig. 2). Based on the structural classification of the CLRs, the receptors included in the Dectin-1 cluster are part of the group V, lacking typical $Ca^{2+}$ and carbohydrate-binding motifs in the CTLD. On the contrary, the receptors included in the Dectin-2 cluster, are part of the structural group II of CLRs, where the CTLD contains both $Ca^{2+}$ and carbohydrate-binding activity (Sancho and Reis e Sousa, 2012b). Whereas all the receptors in the Dectin-1 cluster contain one or more functional motifs within their intracellular tails that allow them to induce intracellular signaling on its own, members of the Dectin-2 cluster (with the exception of

Medical Research Council (MRC) Centre for Medical Mycology, University of Exeter, Exeter, UK. ✉E-mail: m.g.malamud@exeter.ac.uk; gordon.brown@exeter.ac.uk

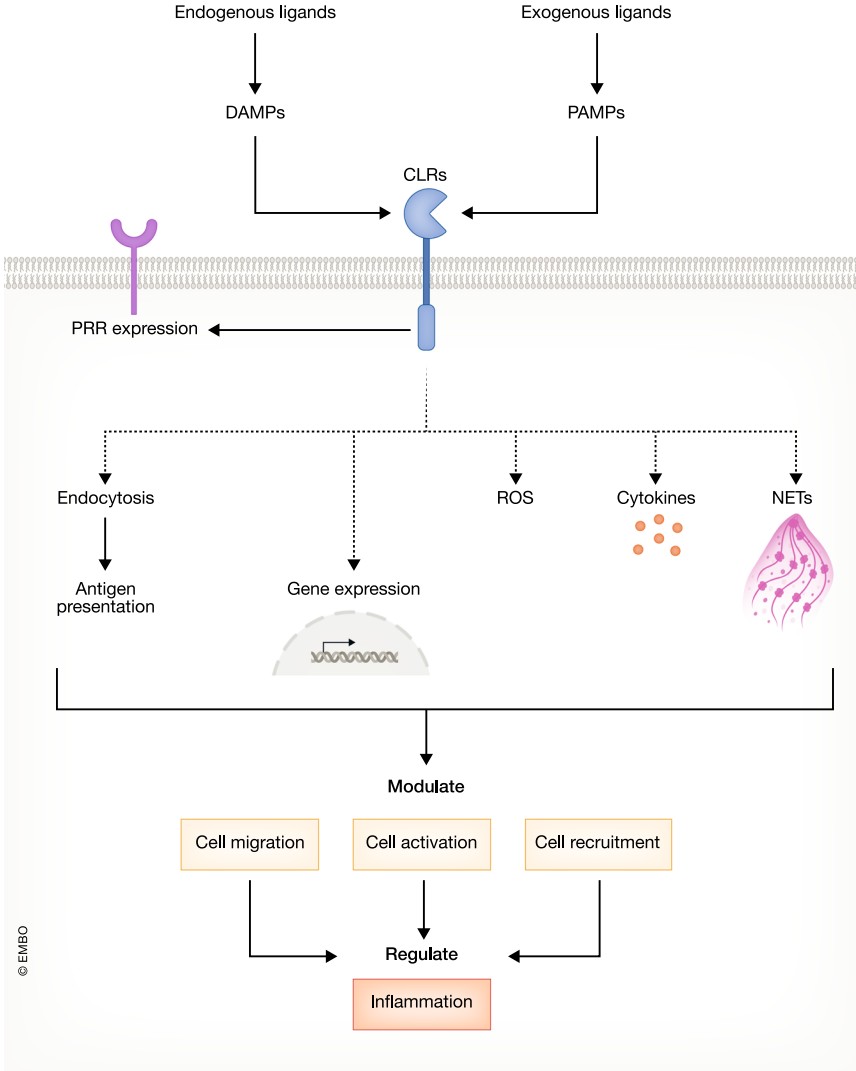

**Figure 1.  Cellular functions of CLRs from the Dectin-1 and Dectin-2 clusters.**

These receptors interact with both microbial components, known as pathogen-associated molecular patterns (PAMPs), and endogenous host-derived molecules, known as damage-associated molecular patterns (DAMPs). Upon recognition of their ligands, CLRs can positively and negatively regulate a wide range of cellular functions such as phagocytosis, chemokine and cytokine production, respiratory burst (ROS) and extracellular trap (NET) formation. In addition, CLR signaling also modifies the expression levels of other pattern recognition receptors (PRRs) and regulates adaptive immune responses. Finally, and based on the combination of signaling of the receptors involved, these responses tailor cellular inflammation.

DCIR) induce intracellular signaling through signaling adaptors, such as the FcRγ (Fig. 2) (Kerscher et al, 2013).

In this review, we will focus on each receptor within both clusters individually and discuss recent discoveries. We will examine their role in host defense from infection, as well as their role in homeostasis, autoimmunity and the recognition of dead cells, detailing expression, ligand nature, signaling pathways, and immune function (Table 1 and Fig. 4).

## The Dectin-1 cluster

### Dectin-1 (CLEC7A)

Originally named for its expression on Dendritic cells (Dendritic cell-associated C-type lectin 1), this receptor is also expressed on

other immune cells, including monocytes, macrophages and neutrophils in both mice and humans, B cells, eosinophils, mast cells and Langerhans cells in humans and Kupffer cells in mice (Mata-Martínez et al, 2022). In addition, single-cell RNA sequencing has showed that Dectin-1 can be also expressed on residential macrophages of the central nervous system under specific conditions, being a key feature of the disease-associated microglia phenotype identified in different types of neuropathologies (Deerhake et al, 2021).

Dectin-1 recognizes β-glucans, a carbohydrate component of cell walls of fungi and plants, and an unidentified ligand from mycobacteria and *Leishmania* (Brown and Gordon, 2001; Lima-Junior et al, 2017; Rothfuchs et al, 2007). Dectin-1 has also been reported to recognize endogenous ligands such as vimentin,

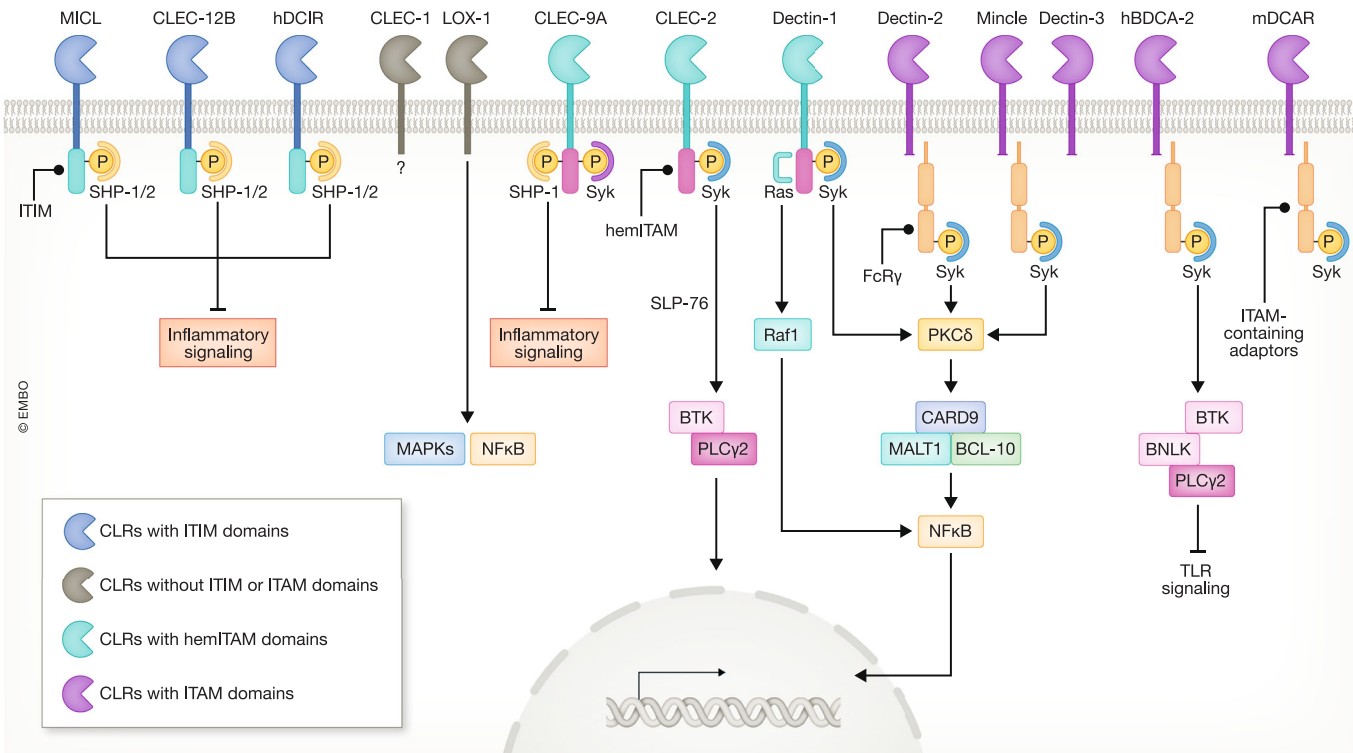

**Figure 2. General signaling pathway of CLRs from Dectin-1 and Dectin-2 clusters.**

The receptors are clustered based on their intracellular signaling domains into CLRs with immunoreceptor tyrosine-based inhibition motif (ITIM) domains, CLRs without typical signaling motifs, CLRs containing hemi immunoreceptor tyrosine-based activating motif (hemITAM) domains or ITAM-coupled CLRs. SHP Src homology region 2 domain-containing phosphatase, MAPKs mitogen-activated protein kinases, NFκB nuclear Factor Kappa B, BTK Bruton's tyrosine kinase, PLCγ2 phosphatidylinositol-specific phospholipase Cγ2, PKC protein kinase C, CARD9 caspase-recruitment domain protein 9, MALT1 mucosa-associated lymphoid tissue lymphoma translocation protein 1, BCL-10 B-cell lymphoma/leukemia 10, BNLK B-cell linker protein.

galactosylated immunoglobulins, galectins, annexins exposed on apoptotic cells, human thioredoxin (hTrx) secreted from cells upon stress and glycans associated with intestinal mucins (Thiagarajan et al, 2013; Daley et al, 2017; Chiba et al, 2014; Roesner et al, 2019; Bode et al, 2019; Shan et al, 2013).

Since the discovery of Dectin-1 as the receptor of β-1,3-glucans in 2001 (Brown and Gordon, 2001), this receptor has become one of the most studied CLRs, specifically its intracellular signaling pathway. The intracellular tail of Dectin-1 contains a conserved tri-acidic sequence (DEDG) followed by a hemITAM motif, and after ligand binding, Dectin-1 signaling occurs through both Syk-dependent and Syk-independent pathways, triggering a variety of cellular responses (Rogers et al, 2005). Recently, different coreceptors were described as required for the recognition of Dectin-1 ligands and the optimal Syk phosphorylation, such as the tyrosine kinase receptor EPH receptor B2 (EPHB2) and the macrophage tetraspanin MS4A4A (Sun et al, 2021; Mattiola et al, 2019). Although previous research established that the intracellular signaling of Dectin-1 requires receptor clustering to form a 'phagocytic synapse,' leading to the exclusion of regulatory tyrosine phosphatases (Goodridge et al, 2011), more recent studies using super-resolution single-molecule localization microscopy have shown that Dectin-1 forms nanoclusters upon stimulation, which can directly trigger intracellular signaling (Li et al, 2022).

One of the major outcomes of Dectin-1 signaling is the Syk-dependent NF-κB activation, which can be induced through canonical and noncanonical pathways. The pro-inflammatory program following NF-κB activation on myeloid cells induces the maturation of dendritic cells and the secretion of cytokines such as IL-1β, IL-2, IL-6, IL-10, IL-23 and TNF-α, helping to initiate an adaptive immune response that involves Th1 and Th17 components, CD8[+] cytotoxic T cells and antibody responses (Gringhuis et al, 2009; LeibundGut-Landmann et al, 2007). Moreover, Dectin-1 activates a Syk-independent pathway mediated by Raf-1 activation, a serine-threonine kinase activated by Ras, that also leads to the activation of NF-κB (Geijtenbeek and Gringhuis, 2009). This pathway also increases IL-12p70 production by human DCs and favors induction of Th1 responses downstream of Dectin-1 (Gringhuis et al, 2009; Sancho and Reis e Sousa, 2012a).

Besides the regulation of transcriptional responses, Dectin-1 signaling can induce different cellular responses. For instance, ligand binding to Dectin-1 triggers actin-mediated phagocytosis, maturation of phagosomes, the respiratory burst, and inflammasome activation (Tone et al, 2019). Moreover, Dectin-1 activation can also trigger an epigenetic reprogramming of monocytes through an Akt–mammalian target of rapamycin (mTOR)–hypoxia-inducible factor-1α (HIF1α) pathway that induce aerobic glycolysis on trained monocytes (Quintin et al, 2012; Cheng et al, 2014).

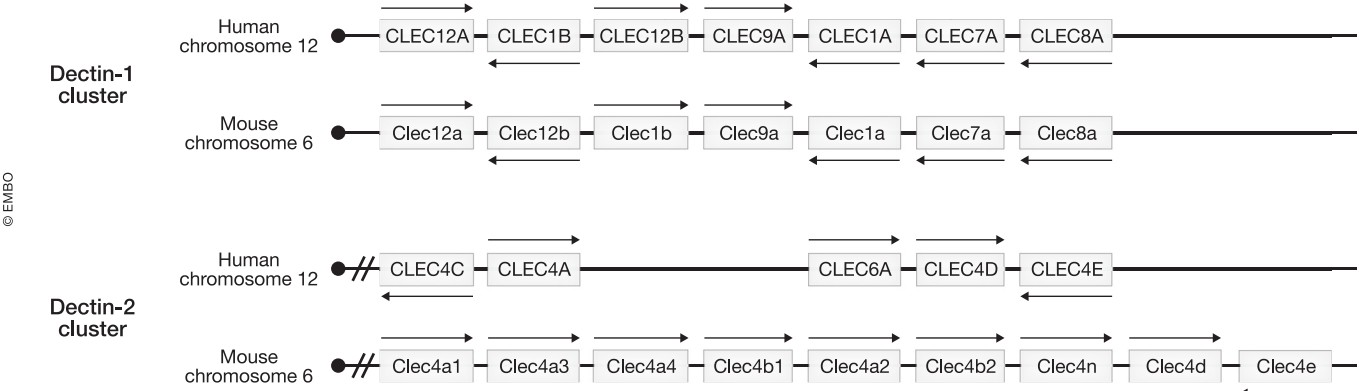

**Figure 3. Genomic organization of CLRs belonging to Dectin-1 and Dectin-2 clusters.**

The Dectin-1 cluster (CLEC12A, MICL; CLEC1B, CLEC-2; CLEC12B, MAH; CLEC9A, DNGR-1; CLEC1A, MelLec; CLEC7A, Dectin-1; CLEC8A, LOX-1) is located in the centromeric part of the Natural Killer gene complex (NK) in both human chromosome 12 in humans and mouse chromosome 6, while the Dectin-2 cluster (CLEC4C, BDCA-2; CLEC4A, DCIR; Clec4b, DCAR; CLEC6A, Dectin-2; CLED4D, MCL; CLEC4E, Mincle) is encoded at the telomeric end of the NK gene complex.

Based on its ability to recognize β-glucans, Dectin-1 has been extensively studied in the context of antifungal immunity, and it is required for mounting a protective immune response against different pathogenic species in mouse models. For instance, Dectin-1-deficient mice have defective DCs responses in the mesenteric lymph nodes following a systemic infection with *Candida spp*, which causes death of antigen-specific CD4[+] T cells in the gut (Drummond et al, 2016). Dectin-1 deficiency is also detrimental during pulmonary infection with *Aspergillus fumigatus*, since Dectin-1 knockout mice present a reduced cytokine production that leads to an insufficient lung neutrophil recruitment, uncontrolled *A. fumigatus* growth and ultimately, higher mortality rates compared with wild-type mice (Werner et al, 2009). Dectin-1 also play an important role in mounting an adaptive immune response against *A. fumigatus*, fine-tuning the levels of Th1 and Th17 cell differentiation (Rivera et al, 2011; Gringhuis et al, 2022). In the case of *Pneumocystis carinii*, an important fungal pathogen in HIV-positive individuals, Dectin-1 mediates macrophage internalization and killing of their cysts in vitro, and it is required for protection against pneumocystis infection in vivo (Saijo et al, 2007; Steele et al, 2003). Furthermore, genetic polymorphisms in Dectin-1 are associated with increased susceptibility to fungal disease in humans. For instance, biallelic deleterious mutation in the gene encoding Dectin-1 has been associated with refractory phaeohyphomycosis, a disease caused by dematiaceous fungi (Drummond et al, 2022). In addition, damaging Dectin-1 variants such as Y238X (early stop codon in Dectin-1), are also associated with disseminated coccidioidomycosis, a disease caused by *Coccidioides immitis* and *C. posadasii* (Hsu et al, 2022).

The role of Dectin-1 has also been studied in *Mycobacteria tuberculosis* infection, and in association with TLR-2, is capable of recognize an unknown ligand of mycobacteria triggering a pro-inflammatory response in macrophages and DCs (Yadav and Schorey, 2006; Shin et al, 2008; Romero et al, 2016). However, using an aerosol model of *M. tuberculosis* infection in mice, Dectin-1 deficiency does not modify the survival of the animals (Marakalala et al, 2011).

Dectin-1 also plays an important role in maintaining intestinal homeostasis. For example, in a murine model of chemically induced colitis, Dectin-1 deficiency leads to the exacerbation of the disease, since the recognition of commensal fungi through this receptor is required to regulate immune responses (Iliev et al, 2012). More recently, an independent study showed that Dectin-1-deficient mice are more susceptible to the dextran sulfate sodium (DSS)-colitis model only when mice are colonized with pathogenic fungi. In contrast, Dectin-1 deficiency protects animals free from fungal colonization, a process associated with the expansion of colonic regulatory T cells and higher levels of *Lactobacillus murinus* in the gut (Tang et al, 2015; Kamiya et al, 2018). Furthermore, in steady state, Dectin-1 is also involved in recognition of mucus (MUC2) from the small intestine in a complex with galectin-3 and FcγIIB, inducing tolerogenic responses through the activation of β-catenin, a transcription factor required by gut tolerogenic DCs, and downregulation of NF-κB activity (Shan et al, 2013).

In addition to infections, Dectin-1 is involved in autoimmunity and allergy, playing both protective and pathogenic functions depending on the context (Deerhake and Shinohara, 2021). For example, in a mouse model of experimental autoimmune encephalomyelitis (EAE), Dectin-1 limits autoimmune neuroinflammation (Deerhake et al, 2021). In this model, after recognition of Galectin-9, Dectin-1 triggers a signalization pathway that involves Syk and the transcription factor NFAT (independent of Card9), leading to the upregulation of oncostatin M, an IL-6-family cytokine with neuroprotective functions (Deerhake et al, 2021). In contrast, using the SKG mouse model of autoimmune arthritis (mice genetically prone to develop arthritis), Dectin-1 recognition of zymosan or purified β-glucans such as curdlan exacerbated the disease after a single intraperitoneal injection, through the activation of Dectin-1–expressing antigen presenting cells (Yoshitomi et al, 2005). Dectin-1 is also required for the inhibition of bone remodeling by Immunoglobulin G during arthritis, enhancing monomeric IgG binding to the low-affinity inhibitory FcγRIIb (Seeling et al, 2023). Interestingly, different groups have studied the role of Dectin-1 in mouse models of experimental autoimmune uveitis, but differences in the methodologies led to conflicting results. (Lee et al, 2016a; Stoppelkamp et al, 2015).

Table 1. Recognized ligands of CLRs from Dectin-1 and Dectin-2 clusters.

| Receptor | Expression | Endogenous | Immune effects | Exogenous | Immune effects |
|---|---|---|---|---|---|
| Dectin-1 (CLEC7A) | Human, mouse | Vimentin (Thiagarajan et al, 2013) Galactosyated immunoglobulins (Karsten et al, 2012) Galectin-9 (Daley et al, 2017) Annexins (Bode et al, 2019) Human thioredoxin (Roesner et al, 2019) Glycans associated with MUC2 (Shan et al, 2013) | Induction of $O_2^-$ production in monocytes; Association with FcγRIIB and suppression of inflammatory signaling; Suppression of antitumor immunity; Suppression of T-cell immunogenicity; Secretion of IL-1β and IL-23 by dendritic cells; Tolerogenic responses | β-1,3-glucans (Brown and Gordon, 2001) Tropomyosin (Gour et al, 2018) | Activation of innate and adaptive immune responses; Regulation of epithelial IL-33 secretion |
| MICL (CLEC12A) | Human, mouse | Monosodium urate crystals (Neumann et al, 2014) Dead cells (Neumann et al, 2014) Neutrophil Extracellular Traps (Malamud et al, 2024) | Regulation of neutrophil activation; Regulation of neutrophil activation and tissue inflammation | Hemozoin crystals (Raulf et al, 2019) | Reduction of antigen cross-presentation |
| CLEC-2 (CLEC1B) | Human, mouse | Podoplanin (Christou et al, 2008) | Activation of platelets | Rhodocytin (Shin and Morita, 1998) Fucoidan (Manne et al, 2013) Diesel particles (Alshehri et al, 2015) | Induction of platelet aggregation; Activation of platelets; Activation of platelets |
| MAH (CLEC12B) | Human, mouse | Unknown | | Unknown | |
| CLEC9A (DNGR-1) | Human, mouse | F-actin (Ahrens et al, 2012) | Induction of antigen cross-presentation to CD8+ T cells | | |
| CLEC-1 (MelLec) | Human, mouse | Histidine-Rich Glycoprotein (Gao et al, 2020) | Suppression of inflammatory responses | DHN-melanin (Stappers et al, 2018) | Protection against systemic A. fumigatus infection |
| LOX-1 (CLEC8A) | Human, mouse | oxLDL (Sawamura et al, 1997) C-reactive protein (Shih et al, 2009) | Induction of endothelial dysfunction; Secretion of IL-8, ICAM-1, and VCAM-1 | GroEL from Klebsiella pneumonia and Escherichia coli (Cahill et al, 2015; Zhu et al, 2013a) | Pyroptosis and activation of macrophages |
| BDCA-2 (CLEC4C) | Human | Serum glycoproteins (Kim et al, 2018) | Reduction of type I interferons secretion | HIV-1 glycoprotein gp120 (Martinelli et al, 2007) Hepatitis C virus glycoprotein E2 (Florentin et al, 2012) Zika Virus non-structural protein 1 (Bos et al, 2020) | Reduction of pDCs activation; Reduction of type I interferons secretion; Reduction of IFN-α secretion |
| DCAR (Clec4b1/2) | Mouse | | | M. tuberculosis Phosphatidylinositol mannosides (Omahdi et al, 2020) Cholesteryl phosphatidyl α-glucoside (Nagata et al, 2021) | Promotion of protective immune responses; Macrophage activation |
| hDCIR (CLEC4A) mDCIR1 (Clec4a2), mDCIR2 (Clec4a4), mDCIR3 (Clec4a3) mDCIR4(Clec4a1) | Human Mouse | Asialo-biantennary N-glycan(s) (Kaifu et al, 2021) Mannotriose, sulfo-Lewis(a), Lewis(b) and Lewis(a) carbohydrates (Bloem et al, 2014, 2013) | Regulation of autoimmune responses; Not assessed | HIV-1 glycoprotein gp120 (Bloem et al, 2014; Lambert et al, 2010) Hepatitis C virus glycoprotein E2 (Florentin et al, 2012) | Promotion of viral replication; Inhibition of IFN production |

**Table 1.** (continued)

| Receptor | Expression | Endogenous | Immune effects | Exogenous | Immune effects |
|---|---|---|---|---|---|
| Dectin-2 (hCLEC6A mClec4n) | Human, mouse | β-glucuronidase (Mori et al, 2017); MUC2 (Leclaire et al, 2018) | Not assessed; Not assessed | α-mannans (McGreal et al, 2006; Saijo et al, 2010); Blastomyces dermatitidis glycoprotein Eng2 (Wang et al, 2017); M. tuberculosis mannose-capped lipoarabinomannan (Yonekawa et al, 2014); House dust mite allergens (Barrett et al, 2009) | Activation of innate and adaptive immune responses; Activation of adaptive responses; Protection against disease; Promotion of airway inflammation |
| MCL (Dectin-3, CLEC4D) | Human, mouse | | | M.tuberculosis trehalose-6,6'-dimycolate (TDM) (Furukawa et al, 2013); α-mannans (Zhu et al, 2013b) | Not assessed; Regulation of gut homeostasis in a model of colitis |
| Mincle (CLEC4E) | Human, mouse | Spliceosome-associated protein (SAP) 130 (Yamasaki et al, 2008a); β-glucosylceramide (β-GlcCer) (Nagata et al, 2017); Cholesterol crystals (Kiyotake et al, 2015); Cholesterol sulfate (Kostarnoy et al, 2017) | Production of inflammatory cytokines; Production of inflammatory cytokines; Production of inflammatory cytokines; Induction of skin inflammatory response | TDM (Ishikawa et al, 2009); TBM (Schoenen et al, 2010); M.tuberculosis glycerol monomycolate (Hattori et al, 2014); Streptococcus pneumoniae glucosyl-diacylglycerol (Behler-Janbeck et al, 2016); Lactiplantibacillus plantarum α-glucosyl diglyceride (Shah et al, 2016); Lentilactobacillus kefiri and Levilactobacillus brevis S-layer glycoproteins (Malamud et al, 2019; Prado Acosta et al, 2021) | Production of inflammatory cytokines and nitric oxide; Production of inflammatory cytokines; Protection of focal pneumonia against S.pneumoniae; Not assessed; Induction of macrophage activation |

In mouse models of allergy induced by repetitive *A. fumigatus* conidia exposure, Dectin-1 deficiency improves lung function. In this model, β-glucan recognition by Dectin-1 enhances the production of pro-inflammatory and proallergic modulators that compromises lung function (Lilly et al, 2012). In the ovalbumin (OVA)-induced airway inflammation model, the absence of Dectin-1 attenuated the disease as a result of increased number of T-regulatory cells in the lungs, mesenteric lymph nodes and the colonic lamina propria, in a process regulated by intestinal commensal microbiota (Han et al, 2021). In contrast, Dectin-1 has a protective function using the dust mite tropomyosin-mediated allergic asthma model in both mouse and non-human primates. In this model, Dectin-1 recognition of tropomyosin, a ubiquitous arthropod-derived molecule, regulates epithelial IL-33 secretion and reduces lung inflammation (Gour et al, 2018).

Dectin-1 has also been implicated in cancer, although it can promote or protect against disease depending on the mouse model of cancer used and the ligand that interacts with. For example, in a mouse model of pancreatic ductal adenocarcinoma (PDA), Galectin-9 binds to Dectin-1 expressed on macrophages, supressing T-cell immunogenicity and accelerating disease progression (Daley et al, 2017). In contrast, activation of Dectin-1 through systemic β-glucan administration, in combination with CD40 agonist antibody therapy was able to eliminate established tumors in a PDA mouse model, demonstrating a protective role for Dectin-1 (Wattenberg et al, 2023). In line with this, during a lung metastasis model of B16F1 melanoma cells, Dectin-1 recognition by macrophages and DCs of N-glycan structures expressed on tumor cells, promoted the activation of the tumoricidal activities of NK cells, controlling disease progression (Chiba et al, 2014). Using the same melanoma cell lines, Zhao et al showed that Dectin-1-activated dendritic cells also promote the differentiation of naive CD4 + T cells to Th9, inducing antitumor responses and controlling disease (Zhao et al, 2016).

Chronic alcohol administration increases the translocation of fungal β-glucan into systemic circulation in mice, and its recognition by Dectin-1-expressing Kupffer cells induce liver inflammation (Yang et al, 2017). In contrast, through the down-regulation of TLR4 signaling, Dectin-1 protects against liver fibrosis in LPS-induced sepsis model (Seifert et al, 2015). More recently, it has been proposed that Dectin-1 recognizes a self-ligand from mouse liver contributing to hepatic exacerbation of inflammation (Torigoe et al, 2024). Thus, depending on the context, Dectin-1 activation can triggers different responses in the liver.

Taking altogether, Dectin-1 is involved in the regulation of many cellular and immunological responses. The ability to recognize multiple ligands, from microbial β-glucan polysaccharides to endogenous DAMPs such as galectin-9 and tumor-associated N-glycans, arguably makes Dectin-1 one of the most versatile myeloid receptors involved in physiological mechanisms.

### MICL (CLEC12A)

MICL (Myeloid inhibitory C-type lectin-like), also known as CLL-1, DCAL-2 and KLRL-1, is primarily expressed on myeloid cells, such as monocytes, macrophages, polymorphonuclear cells and dendritic cells in both humans and in mice, but only on B cells in mice (Marshall et al, 2004). It is composed of a single CTLD that lacks the necessary residues for calcium-binding and an ITIM-bearing cytoplasmic tail. Recently, was shown that the Cysteine

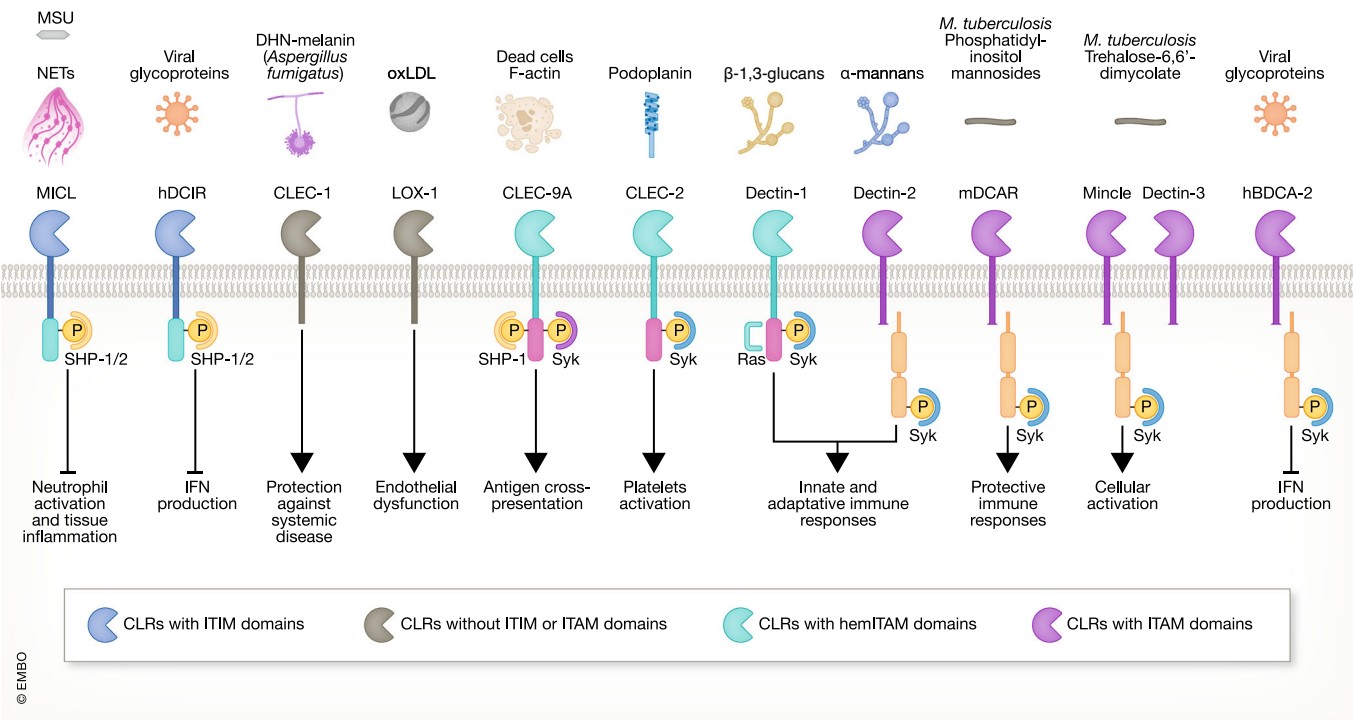

**Figure 4. CLRs-ligand interactions.**

Selected ligands of Dectin-1 and Dectin-2 clusters are illustrated, with receptors color-coded according to their intracellular signaling domains. The main responses induced by the interaction between each ligand and receptor are also highlighted.

118 present on the stalk region plays a key role in MICL cell-surface expression (Vitry et al, 2021).

After ligand recognition, the ITIM motif is phosphorylated and recruits SHP-1 and SHP-2, which negatively regulates inflammatory cellular responses. Antibody cross-linking on human neutrophils phosphorylates the MICL ITIM motif in a flotilin-rich membrane domain in a Src-dependent manner (Paré et al, 2021). Recently, it has been proposed that after ligand stimulation, CLEC12A ITIM motif is dispensable for signaling, but the transmembrane region of the receptor disrupts lipid raft recruitment induced by specific agonists attenuating intracellular signals (Xu et al, 2023). Therefore, the intracellular signaling pathway of MICL is still not completely understood (Fig. 2).

MICL recognizes endogenous ligands present on the surface of cells isolated from heart, lungs, liver, spleen and kidney, but the nature of these structures has not been identified yet (Py et al, 2008). MICL also recognizes dead cells, and monosodium urate crystals (MSU), which are key danger signals for cell-death-induced immunity and consequently, MICL-deficient mice exhibit hyperinflammatory responses to MSU or necrotic cells (Neumann et al, 2014). MICL regulates immune cellular responses during the collagen antibody-induced arthritis model, where MICL-deficient mice present an exacerbated disease that was also reproduced by administering MICL-blocking antibodies in wild-type mice (Redelinghuys et al, 2016). MICL also recognizes an unknown ligand on endothelial cells which facilitates the binding and transmigration of DCs across the blood–brain barrier, and both antibody targeting and the genetic deletion of MICL protect animals in a mouse model of experimental autoimmune encephalomyelitis (Sagar

et al, 2017). Recently, we have discovered that this receptor directly recognizes neutrophil extracellular traps (NETs), and that this interaction is essential to regulate neutrophil activation (Malamud et al, 2024). Importantly, patients suffering from inflammatory conditions in which NETs are linked to disease pathology, such as RA, systemic lupus erythematosus and severe COVID-19 presented antibodies targeting MICL capable of blocking the function of this receptor (Malamud et al, 2024). Thus, MICL represents a universal, novel auto-regulatory pathway that helps prevent aberrant neutrophil activation and the resulting tissue damage in inflammatory conditions (Malamud et al, 2024).

The biological function of this receptor can change during the course of infection, triggering protective responses or promoting disease depending on the microorganism encountered. MICL is as a receptor for hemozoin crystals, a *Plasmodium*-derived product, and contributes to the progression of disease in a mouse model of experimental cerebral malaria (Raulf et al, 2019). The interaction between hemozoin crystals and MICL on BMDCs affects the cross-presentation of plasmodial antigens to CD8$^+$ T cells, and consequently MICL-deficient mice are more protected (Raulf et al 2019). In line with these results, during a lymphocytic choriomeningitis virus infection, MICL amplifies the signals elicited by the RNA sensor RIG-I, increasing IFN-I responses via Src family tyrosine kinases and consequently, MICL deficiency protects animals from disease (Li et al, 2019). In contrast, MICL play an important regulatory role in antibacterial autophagy through a functional interaction with an E3- ubiquitin ligase complex, and consequently MICL-deficient mice are more

susceptible to Salmo*nella* infection in vivo (Begun et al, 2015). MICL also binds to *Legionella pneumophila*-derived ligands, but does not play any role in innate immune responses against this bacterium (Klatt et al, 2023). MICL also recognizes mycolic acids from different mycobacterium species, regulating host immune responses during infection (Nishimura et al, 2023).

Human MICL is important in the context of acute myeloid leukemia (AML), since the expression of this receptor, together with CD123, represent a strong prognostic marker for leukemia relapse (Roug et al, 2014). Moreover, the surface expression of MICL is upregulated in leukemic stem cells compared to the normal hematopoietic stem cells (HSCs), making this receptor a potential target in human AML (Williams et al, 2019). In fact, different immunotherapies targeting MICL have been developed, such as chimeric antigen receptor T cells (CAR-Ts) or bispecific antibody targeting (Laborda et al, 2017; van Loo et al, 2019). For example, two independent studies demonstrated that CAR-Ts specific for MICL exhibit potent cytotoxicity against MICL-expressing AML cell lines and primary AML samples without disrupting normal HSCs, and extend animal survival in a human xenograft mouse model (Tashiro et al, 2017; Laborda et al, 2017).

MICL, as an inhibitory receptor, can downregulate cellular responses, though the details of its intracellular signaling pathway remain largely unknown. Gaining a deeper understanding of the molecular mechanisms behind its inhibitory function could reveal promising therapeutic opportunities. For example, activating MICL signaling with specific antibodies might help alleviate symptoms in inflammatory conditions. Conversely, blocking MICL functionality could be advantageous during infections by preventing its interaction with DNA from neutrophil extracellular traps (NETs) in systemic fungal infections, or with hemozoin crystals during *P. falciparum* infection. Importantly, these antibodies must be carefully engineered to achieve the desired therapeutic effect.

### CLEC-2 (CLEC1B)

CLEC-2 is a type II transmembrane receptor that contains a hemITAM motif in its intracellular domain. Originally identified through its interaction with rhodocytin, a heterodimeric $(\alpha\beta)_2$ C-type lectin toxin isolated from the venom of *Calloselasma rhodostoma*, this receptor is expressed on various myeloid cells, including neutrophils, macrophages, activated monocytes, and distinct subsets of dendritic cells, and it is also highly expressed on megakaryocytes and platelets (Suzuki-Inoue et al, 2006; Kerrigan et al, 2009; Senis et al, 2007). The receptor exists as both monomer and homo-dimer on the surface of resting platelets, a process dependent of N-glycosylation levels of the stalk region of the monomers (Hughes et al, 2010; Pollitt et al, 2014). After ligand recognition, CLEC-2 undergoes multimerization, a process that clusters together several hemITAMs motifs, and triggers the phosphorylation of the SH2 domains in a Syk-dependent manner (Fig. 2). This process activates an intracellular signaling pathway that involves the adaptor protein SLP-76, the activation of Bruton tyrosine kinase and PLCγ2 (Hughes et al, 2010).

CLEC-2 recognizes podoplanin, a mucin-type transmembrane glycoprotein expressed on lymphatic endothelial cells (Suzuki-Inoue et al, 2007; Christou et al, 2008). This interaction is required for lung development, the separation between lymphatics and blood vessels and preservation of lymph node vascular integrity (Bertozzi et al, 2010; Suzuki-Inoue et al, 2010; Herzog et al, 2013).

In this sense, a recent study has suggested that a dysfunctional CLEC-2 could be associated with the development of Gorham-Stout disease, a lymphangiomatosis characterized by abnormalities on the lymphatic vessels distribution (Oishi et al, 2024). Moreover, through podoplanin binding, CLEC-2 mediates thrombosis and wound repair, being particularly important under inflammatory conditions where podoplanin is upregulated on stromal cells and macrophages (Rayes et al, 2019). In addition, it has been suggested that CLEC-2 interaction with podoplanin reduces the accumulation of inflammatory macrophages in the mouse peritoneum after the challenge with LPS, accelerating the cell migration to mesenteric lymph nodes (Bourne et al, 2021). A recent study indicates that another endogenous ligand for CLEC-2 is human Dectin-1, an interaction mediated by an O-glycosylated motif present in the stalk region of Dectin-1 (Haji et al, 2022) (see Box 1).

Considering the high expression of CLEC-2 in platelets, several studies have analyzed the role of this receptor in the inflammation during atherosclerosis and thrombosis and have led to the discovery of new potential CLEC-2 ligands. For example, the smooth muscle calcium-binding protein S100A13, a protein that is exposed after injuries to vascular endothelium, activates platelets in a process mediated by CLEC-2 but independent of podoplanin, suggesting that S100A13 could be an endogenous ligand of CLEC-2 (Inoue et al, 2015; Meng et al, 2021). In addition, hemin, an oxidized form of the group heme, induces platelet aggregation and that process is significantly reduced in CLEC-2-deficient platelets (Bourne et al, 2020). Supporting this result, a previous study showed that protoporphyrin IX, the precursor of heme, and cobalt hematoporphyrin also bind to CLEC-2 and inhibit podoplanin-CLEC-2 interactions without inducing platelet activation, reinforcing the idea of an interaction between CLEC-2 and porphyrins (Tsukiji et al, 2018).

---

**Box 1  Dectin-1 and Dectin-2 clusters crosstalk**

CLRs typically encounter ligands in complex structures that can simultaneously bind multiple CLRs and/or PRRs, with the combined signaling shaping the overall immune response (Del Fresno et al, 2018). Recent evidence suggests that crosstalk between Dectin-1 and Dectin-2 receptor clusters occurs in response to ligands recognized by two CLRs, with the integration of these signals determining the final immune outcome. For example, it has been shown that human thioredoxin interacts with both Dectin-1 and Dectin-2, inducing IL-23 through Dectin-1 binding and IL-1β via either Dectin-1 or Dectin-2, highlighting how different intracellular signaling pathways impact the cellular response to the same ligand (Roesner et al, 2019). Similarly, Dectin-1 has been shown to interact with Dectin-2, activating the NLRP3 inflammasome in response to *Histoplasma capsulatum* (Chang et al, 2017). In contrast, recognition of *Fonsecaea monophora*, a pathogenic fungus responsible for chromoblastomycosis, by Mincle suppresses Dectin-1 and Dectin-2 responses, underscoring how cooperation between CLR signaling pathways influences the immune response (Wevers et al, 2014). More recently, Dectin-1 was found to serve as a ligand for CLEC-2, highlighting the potential for CLRs to modulate immune responses through heterophilic interactions (Haji et al, 2022). In addition, some CLRs form heterodimeric receptors, such as MCL and Mincle, where each partner is required for the surface expression of the other, or MCL and Dectin-2, which cooperate when recognizing a shared ligand. Understanding how CLRs within the Dectin-1 and Dectin-2 clusters interact and regulate signaling, whether through recognition of the same ligand or crosstalk between receptors, will be essential to fully appreciate their roles in homeostasis and host-pathogen immunity.

Plasma levels of the soluble form of CLEC-2, generated by protease cleavage, are risk factors for coronary artery disease, and prognostic indicators of vascular events in patients with acute ischemic stroke and cancer (Fei et al, 2020; Inoue et al, 2019; Wu et al, 2019b; Xiong et al, 2016). In cancer, although the interaction between CLEC-2 and podoplanin-expressing tumor cells promote angiogenesis, tumor growth and metastasis, the expression of this receptor in gastric cancer cells suppresses metastasis (Wang et al, 2016a; Kato et al, 2007). Using small-hairpin RNAs to knock down CLEC-2 expression in gastric cancer cell lines, the injection of CLEC-2-deficient cells form more metastases in mice compared with CLEC-2-expressing cells, in a Syk-dependent manner, although the mechanism involved is unclear (Wang et al, 2016a).

Highly expressed on platelets, CLEC-2 plays a crucial role in maintaining vascular integrity. By recognizing podoplanin, CLEC-2 mediates platelet activation, and blocking this interaction has been proposed as a potential target for regulating thrombosis. In this context, a recent study demonstrated that specific structural modifications to rhodocytin can convert it into an antagonist, preventing podoplanin from binding to CLEC-2 and thereby inhibiting platelet activation (Obermann et al, 2024). As such, identifying novel strategies to block CLEC-2 activation will be key in developing new therapeutic agents.

### CLEC12B (macrophage antigen H: MAH)

Perhaps one of the least well studied receptors in the Dectin-1 cluster, CLEC12B, was discovered based on the homology with NKG2D, a C-type lectin-like receptor present on NK cells (Hoffmann et al, 2007). It has been detected at the protein level on the human monocyte cell line, U937, following stimulation with phorbol 12-myristate 13- acetate, and recently on skin mast cells (Iijima et al, 2021). CLEC12B contains an ITIM motif in its intracellular domain able to recruit SHP-1 and SHP-2 and inhibit ITAM-based signaling (Fig. 2). Until now, there are no ligands that have been discovered for this receptor, but it has been suggested that interacts with caveolin-1, a small scaffolding protein (Kulkarni et al, 2013). Based on proteomic data using mouse embryonic fibroblasts, CLEC12B is upregulated in Caveolin-1-deficient mice, suggesting a possible interaction between these two proteins (Kulkarni et al, 2013).

Based on the presence of polymorphism in CLEC12B linked to one family with predisposition to childhood cancer it has been proposed as a candidate cancer predisposition gene (Derpoorter et al, 2019). On the other hand, a recent study showed that CLEC12B inhibits tumor growth in lung cancer (Chi et al, 2021). CLEC12B impairs cell proliferation, enhances cell apoptosis and inactivates the PI3K/AKT signaling, in a process dependent on CLEC12B-SHP-1 interaction (Chi et al, 2021). Moreover, CLEC12B overexpression increased SHP-1 level, suggestion a coregulation of the receptor and the phosphatase (Chi et al, 2021). CLEC12B together with other inhibitory receptors, such as MICL, are upregulated in Behçet's syndrome, an autoinflammatory disorder characterized by blood vessel inflammation and an exaggerated innate immune response, suggesting that these receptors could be an alternative therapeutic target in the control of the disease (Oğuz et al, 2016).

So far, CLEC12B is an orphan receptor, and the identification of both exogenous and endogenous ligands will provide much information into the physiological role of this receptor. In addition,

the high degree of sequence homology between species suggests that this receptor plays a key real function yet to be discovered.

### CLEC9A (DNGR-1)

The expression pattern of CLEC9A is mainly restricted to type 1 conventional dendritic cells (cDC1s) in both mice and humans. In mice, it is also expressed on DC progenitors and in a lower extent, on plasmacytoid DCs (pDCs). However, in humans it is not expressed on pDCs, being only expressed by immature BDCA3$^+$ DCs and on a small subset on CD14$^+$CD16$^-$ monocytes (Sancho et al, 2009, 2008; Poulin et al, 2012; Huysamen et al, 2008; Caminschi et al, 2008). Moreover, this characteristic expression profile on DCs has promoted its use both as a cellular marker and as DC lineage tracer (Schraml et al, 2013; Tone et al, 2019).

Dimerization of CLEC9A via cysteine residues located in the neck region of this receptor is critical for its functionality (Hanč et al, 2016). Although the intracellular domain contains a hemITAM motif that after ligand binding induces signaling through Syk kinase, CLEC9A does not induce cellular activation via NFκB. Instead, CLEC9A induce antigen cross-presentation to CD8$^+$T cells through diversion of cargo to endosomal compartments (Fig. 2) (Sancho et al, 2009; Zelenay et al, 2012; Iborra et al, 2012). Importantly, ligand recognition of CLEC9A leads to rapid activation of CBL and CBL-B E3 ligases that cause Syk ubiquitination, terminating signaling and limiting antigen cross-presentation (Henry et al, 2023).

The only recognized ligand for CLEC9A is Filamentous actin (F-actin), an intracellular component that is exposed when the membrane integrity is lost (Zhang et al, 2012; Ahrens et al, 2012). This interaction is enhanced by the presence of myosin II, an F-actin-associated motor protein, and inhibited by secreted gelsolin, an extracellular actin-binding protein (Schulz et al, 2018; Giampazolias et al, 2021). CLEC9A signaling in phagosomes containing necrotic cells-derived antigens induce the rupture of the phagosome membrane through SYK and NADPH oxidase activation, leading to the release of antigenic material into the cytosol of cDC1s, where they can enter the endogenous MHC class I presentation pathway (Canton et al, 2021). In line with this finding, CLEC9A cross-presentation ability is required for promoting protective CD8$^+$ T-cell responses to vaccinia virus or herpes simplex virus infection (Zelenay et al, 2012; Iborra et al, 2012). On the contrary, CLEC9A is not required for protection during respiratory syncytial virus infection, where direct presentation by DCs initiate CD8 + T-cell responses (Durant et al, 2014).

The restricted expression profile on cDCs together with the ability to induce cross-presentation make CLEC9A a receptor of interest in the development of antigen-targeting strategies to increase the efficacy of cancer immunotherapies and vaccines (Zeng et al, 2018; Tullett et al, 2016). For example, antibody targeting of CLEC9A not only induces CD8 T-cell responses, but also promotes MHC-II antigen presentation to CD4 + T cells and antibody responses (Caminschi et al, 2008; Li et al, 2015; Joffre et al, 2010). This system has also been used to deliver antigen-containing nanoemulsions with immunostimulatory properties. Different targeting strategies have also been tested, such as beads coated with a synthetic F-actin/myosin II complex or specific small peptides discovered through in silico approaches (Zeng et al, 2018; Cueto et al, 2020).

CLEC9A recognition of cell death in a mouse model of acute pancreatitis or after systemic *C. albicans* infection inhibited the

production of the neutrophil-recruiting chemokine MIP-2 by cDC1s, reducing neutrophil recruitment and promoting disease tolerance (del Fresno et al, 2018). This process is mediated by the recruitment of the inhibitory phosphatase SHP-1 to the cytoplasmic tail of CLEC9A, which in turns downregulates NF-κB activation triggered by heterologous receptors on cDC1s (del Fresno et al, 2018). Similarly, CLEC9A expressed on macrophages also regulates in vitro neutrophil recruitment and activation in response to heat-killed *M. tuberculosis*, a process dependent on the levels of IL-1β and CXCL8 secretion (Cheng et al, 2017). On the contrary, CLEC9A promotes inflammation in mouse models of atherosclerosis (Haddad et al, 2017). In this case, deletion of CLEC9A in both Ldlr$^{-/-}$ and Apoe$^{-/-}$ mouse models promotes an anti-inflammatory and antiatherogenic response with an increase of *Il10* and *Tgfb*, reducing macrophage and T-cell infiltration within the plaques (Haddad et al, 2017).

CLEC9A plays an important role in the regulation of immune responses against dead cells, when it interacts with exposed F-actin. Targeting CLEC9A has proven to be an attractive strategy to enhance tumor immunogenicity and the development of new approaches to utilize the functions of this receptor may have important implications in the modulation of antitumor (and possibly antiviral) immune responses.

### CLEC-1 (MelLec)

CLEC-1 contains a single CTLD and so far, its signaling pathway has not been discovered. Is it known that the intracellular domain does not contain an ITAM or ITIM motif, and although the receptor contains a YSST and tri-acidic DDD motif in its cytoplasmic tail, their involvement in the downstream signaling remains uncharacterized (Fig. 2) (Plato et al, 2013; Lopez Robles et al, 2017a). It is expressed by human, mice and rats endothelial cells and it is also present on humans and rats myeloid cells, including various DC populations, monocytes, macrophages, and granulocytes (Stappers et al, 2021; Lopez Robles et al, 2017a).

CLEC-1 plays a role in antifungal immunity, since this receptor recognize the naphthalene-diol unit of 1,8-dihydroxynaphthalene (DHN)-melanin, a component found in conidial spores of *A. fumigatus* as well as in other DHN-melanized fungi (Stappers et al, 2018). CLEC-1 deficiency led to increased fungal burdens and alterations of the inflammatory responses after systemic infection with *A. fumigatus*, and it was critical for mounting a protective immune response (Stappers et al, 2018). Consistent with this, a single-nucleotide polymorphism in the CLEC-1 intracellular domain was associated with increased susceptibility to disseminated *A. fumigatus* infections in stem-cell transplant patients. On the other hand, CLEC-1 promotes pulmonary allergic inflammation in response to *A. fumigatus* spores in mice (Tone et al, 2021). Although CLEC-1-deficient mice had higher fungal burdens compared to wild-type, the animals did not present apparent adverse effects, suggesting that CLEC-1 is required to control fungal burdens in the lungs, but the inflammatory response elicited by this receptor has a negative impact on the animals (Tone et al, 2021). In this allergic model, it was suggested that the protective effect of CLEC-1 deficiency is consequence of a reduced pulmonary inflammatory response, characterized by a reduced neutrophil influx to the lungs of those animals (Tone et al, 2021). This is consistent with a delayed neutrophil recruitment during the systemic model of *A. fumigatus* infection in CLEC-1-deficient mice

(Stappers et al, 2018). CLEC-1 also downregulates neutrophil recruitment in a mouse model of acute liver injury, restraining inflammatory responses (Ligeron et al, 2024).

CLEC-1 is also involved in adaptive immunity. CLEC-1 recognize dead cells, and regulates antigen cross-presentation by dendritic cells, limiting T-cell responses (Drouin et al, 2022). In rats, CLEC-1 modulates T-cell responses, and the deficiency of the receptor enhance CD4$^+$Th1 and Th17 responses both in vitro and in vivo (Lopez Robles et al, 2017b). Furthermore, in human lung transplants, a decreased CLEC-1 expression was associated with increased levels of IL-17A and chronic rejection (Lopez Robles et al, 2017b; Tone et al, 2019).

In summary, CLEC-1 is involved in the recognition of both exogenous and endogenous ligands (Table 1). Given its role in antifungal immunity, it is important to understand CLEC-1 signaling pathway and how it mediates its biological functions.

### LOX-1 (CLEC8A)

LOX-1 was originally discovered as a membrane scavenger receptor involved in the internalization of oxidized low-density lipoproteins (oxLDL) by endothelial cells, but it is also expressed on other cell types, including smooth muscle cells, neurons, fibroblasts, platelets, and different myeloid cells (Sawamura et al, 1997; Kakutani et al, 2000). Structurally, LOX-1 forms homodimers through conserved cysteine residues present in the extracellular domain (Xie et al, 2004). The recognition of oxLDL, a negatively charge molecule, is attributed to a CTLD terminal cluster of positively charged amino acids (Ohki et al, 2005). LOX-1 also recognizes other ligands such as C-reactive protein (CRP), activated platelets, apoptotic cells, and bacterial and advanced glycation end products (Jin and Cong, 2019; Shih et al, 2009).

After ligand recognition, and through mechanisms not completely understood, LOX-1 activates different downstream pathways with several cellular effects, including ROS production through Rac-mediated NADPH oxidase activation, the expression of chemokines and adhesion molecules through NFκB and the activation of NLRP3 inflammasome and the consequent production of IL-1β (Fig. 2) (Sugimoto et al, 2009; Ding et al, 2014). It has also been suggested that the membrane N-terminal fragments of LOX-1, and their regulation by the signal peptide peptidase-like 2a and b (SPPL2a/b) play an important role in the intracellular signaling of the receptor (Mentrup et al, 2019).

The expression of LOX-1 is low under normal physiological conditions but can be induced in the presence of inflammatory cytokines and oxLDL (Feng et al, 2014; Kattoor et al, 2019). On the contrary, statins, lipid-lowering drugs, are able to reduce LOX-1 expression on endothelial cells (Li et al, 2002; Biocca et al, 2015). Surprisingly, this process is not mediated by statins-induced lowering levels of oxLDL, but instead there is a direct interaction between statins and the hydrophobic portion of LOX-1 that alters the structure of the receptor-binding domain and modifies the oxLDL/Lox-1 axis downstream effects (Kattoor et al, 2019; Matarazzo et al, 2012).

LOX-1 is involved in the pathogenesis of atherosclerosis and associated cardiovascular diseases, such as hypertension and myocardial ischemia (Barreto et al, 2020). Recognition of oxLDL via LOX-1 on epithelial cells induces endothelial dysfunction, a key event in the initiation and progression of atherosclerosis. This process is characterized by persistent inflammation and ROS

production, which in turn activates NF-κB and induces the expression of chemokines and adhesion molecules that facilitates the recruitment of monocytes, plaque formation and proliferation of vascular smooth muscle cells (Celermajer, 1997; Sawamura et al, 1997; Tian et al, 2019). Furthermore, the elevated ROS production promotes the oxidation of LDL to oxLDL, amplifying ROS production through LOX-1. In humans, different studies showed that polymorphisms and alternative splice variants of LOX-1 gene could be associated with either protection or promotion of cardiovascular diseases (Rizzacasa et al, 2017). It has also been proposed that the levels of the soluble form of LOX-1 (sLOX-1), a consequence of the proteolytic action of ADAM10 proteases on cell bound LOX-1, could be used as a candidate for earlier diagnosis and to provide risk estimates of cardiovascular disease development (Mentrup et al, 2019; Hofmann et al, 2020; Inoue et al, 2010; Li et al, 2018; Yokota et al, 2016). A recent study has linked LOX-1 with cardiovascular disease in Covid-19 patients, showing that the high expression of the receptor in humans immature neutrophils (CD10-CD64 + ) infiltrating the bronchoalveolar space in the lungs during infection is strongly associated with a high risk of severe thrombosis (Combadière et al, 2021).

Using LOX-1-deficient mice it has been shown that this receptor is detrimental in animal models of arthritis and osteoarthritis (Hashimoto et al, 2016, 2018). In addition, LOX-1 expression has been detected in the chondrocytes of patients with rheumatoid arthritis (RA) and the stimulation of Human RA fibroblast-like synoviocytes with oxLDL leads to the production of matrix metalloproteinases, enzymes that degrade extracellular matrix (Ishikawa et al, 2012).

LOX-1 has also been implicated in infectious diseases. For instance, macrophage LOX-1 interacts with GroEL, a surface associated protein present on outer membrane vesicles of *Klebsiella pneumonia*, leading to the pyroptosis of macrophages and the release of pro-inflammatory cytokines (Cahill et al, 2015). In addition, GroEL expressed on the surface of *Escherichia coli* is recognized by macrophage LOX-1 leading to the phagocytosis of the pathogen (Zhu et al, 2013a).

Considering all the data, targeting LOX-1 through different modulators such as monoclonal antibodies, the use of statins and by microRNAs has become an interesting strategy in tackling atherosclerosis, cardiovascular diseases and osteoarthritis (Hein et al, 2014; Luo et al, 2016; Hofmann et al, 2018).

## Dectin-2 cluster

### Blood dendritic cell antigen 2 (BDCA-2, CD303, hCLEC4C)

Discovered initially by a monoclonal antibody against CD4[+] blood DCs, BDCA-2 is recognized as a specific marker for human pDCs (Dzionek et al, 2000; Boiocchi et al, 2013). In fact, until now there is no murine homolog described. This receptor does not contain a signaling motif in its intracellular domain and interacts with the adaptor transmembrane protein FcεRIγ to transduce intracellular signals, in a process that involves tyrosine phosphorylation of Syk, Blnk and PLCγ2 activation (Fig. 2) (Cao et al, 2007; Röck et al, 2007). Surprisingly, signaling through BDCA-2 appears to reduce the activation of the NFκB pathway, in particular in response to TLR ligands, inhibiting the production of type I interferons and other cytokines such as tumor necrosis factor (TNF)-related apoptosis-inducing ligand (TRAIL) (Riboldi et al, 2009, 2011).

BDCA-2 signaling also involves the activation of the MEK1/2-ERK pathway, being critically involved in the inhibition of type I interferon production (Janovec et al, 2018).

BDCA-2 stimulation inhibits interferon production, and it has been proposed as a therapeutic target in different pathologies linked to high levels of type I interferon, such as Systemic lupus erythematosus (SLE) (Reily et al, 2019). For instance, even though SLE patients have a reduced number of BDCA-2-expressing pDCs, their IFN alpha production could be inhibited by a monoclonal antibody (mAb) targeting BDCA-2 (Blomberg et al, 2003). In addition, a single dose of a humanized mAb that binds BDCA-2 decreased IFN expression and improved cutaneous lupus disease activity in SLE patients (Chaichian et al, 2019; Furie et al, 2019, 2022). BDCA-2 also binds to IgG through the recognition of the galactose-terminated biantennary glycans, suggesting that normal levels of immunoglobulins are also able to inhibit interferon production (Jégouzo et al, 2015; Kim et al, 2018). Interestingly, SLE is associated with a reduced IgG galactosylation and mechanistically, this could prevent the binding of this molecule to BDCA-2 on pDCs potentially relieving the inhibitory signaling in these cells (Kim et al, 2018).

In contrast, BDCA-2 has a detrimental role during viral infections, where it has been shown that the interaction between this receptor and viral ligands downregulate type I interferons responses. For example, BDCA-2 is able to recognize the HIV-1 envelope glycoprotein gp120 suppressing the activation of pDCs in TLR9-mediated responses (Martinelli et al, 2007). In addition, the hepatitis C virus glycoprotein E2 is also recognized by BDCA-2 and this interaction blocks the TLR7/9-mediated production of type I interferons by pDCs (Florentin et al, 2012). Zika virus is able to reduce the expression of BDCA-2 on pDCs, reducing the secretion of IFN-α and limiting cell activation, in a process mediated by the interaction of BDCA-2 with the Zika Virus non-structural protein 1 (NS1) (Bos et al, 2020). BDCA-2 also interacts with heparin and results in inhibition of TLR9-driven type I IFN production in primary human pDCs (Venegas-Solis et al, 2024).

The ability of BDCA-2 to reduce type I interferon production makes this receptor an interesting drug target in several diseases. Given the dual role of BDCA-2 in SLE and in viral infections, further studies are needed to assess the potential risks and complications of these treatments.

### Dendritic cell immunoactivating receptor (DCAR; Clec4b1/2)

There are two different DCAR described in mice: DCAR (gene symbol Clec4b1), also known as DCAR2, and DCAR1 (gene symbol Clec4b2) (Kanazawa et al, 2003; Kaden et al, 2009). DCAR1 expression is tissue-dependent and restricted to CD8[+] DCs in spleen and thymus and to CD11b[+] subpopulations in spleen and bone marrow (Kaden et al, 2009; Kerscher et al, 2013). DCAR2 is predominantly expressed in monocyte-derived cells from lungs and spleen, and in small peritoneal macrophages (CD11b[+]CD11c[+]CD115[+]MHC class II[hi] population) (Toyonaga et al, 2016). In addition, DCAR2 is highly expressed on a subpopulation of conventional DCs (CD11c[hi]MHC-II[hi] cells), in bone marrow and skin-draining lymph nodes (Kishimoto et al, 2015). Rats also express DCAR, but only the Clec4b2 homolog, which is express in CD4[+] myeloid cells, neutrophils and eosinophils (Aoun et al, 2021; Bäckdahl et al, 2020; Daws et al, 2019).

DCARs signaling pathway is not completely understood (Fig. 2). Although it was suggested that these receptors interact with the

protein adaptor FcRγ chain, this was only demonstrated for DCAR2 where the cross-linking of the receptor in the presence of γ chain activates calcium mobilization and tyrosine phosphorylation of cellular proteins (Kanazawa et al, 2003). Moreover, the expression of surface DCAR2 on small peritoneal macrophages was not detectable in FcRγ$^{-/-}$ mice, indicating a role for FcRγ in the surface expression of this receptor (Toyonaga et al, 2016). In the case of DCAR1, after receptor binding to antigen-anti-DCAR1 complexes, the complexes are internalized and lead to the initiation of Ag-specific T-cell responses. In addition, triggering of DCAR1 modulates DCs function towards a pro-inflammatory response, enhancing the levels of IL-12p70 and downregulating IL-10, suggesting an activatory role for this receptor (Kaden et al, 2009; Kerscher et al, 2013).

DCARs recognizes phosphate-containing phosphatidyl groups, which enables recognition of some bacterial pathogens including mycobacteria and *Helicobacter pylori* (Omahdi et al, 2020). Mycobacterial cell wall contains acylated Phosphatidylinositol mannosides (PIMs) that interact with DCAR on peritoneal macrophages, leading to the production of monocyte chemoattractant protein 1. In addition, following a mycobacterial infection, CLEC4b1-deficient mice displayed reduced numbers of inflammatory monocytes and higher bacterial burdens in their peritoneal cavities, indicating that DCAR promotes protective immune responses (Toyonaga and Yamasaki, 2020). In the case of *H. pylori*, it has been shown that DCAR2 recognizes cholesteryl phosphatidyl α-glucoside, a metabolite formed by bacterial modification of the host cholesterol (Nagata et al, 2021).

DCAR also plays a role in autoimmunity. A single-nucleotide polymorphism in rats regulating the expression of Clec4b2 controls the development of arthritis, since CD4$^+$ cells expressing the receptor are able to limit the expansion of arthritogenic T cells (Bäckdahl et al, 2020). In addition, Clec4b2 controls neutrophil recruitment and activation during early stages of arthritis (Aoun et al, 2021).

In conclusion, DCAR is involved in the regulation of immune responses during infection and autoimmunity. However, an endogenous ligand for this receptor has not yet been identified. The discovery of this molecule(s) will open new perspectives in our understanding of the physiological role of DCAR.

### Dendritic cell immunoreceptor (h: DCIR, CLEC4A; m: DCIR1 (Clec4a2), DCIR2 (Clec4a4), DCIR3 (Clec4a3), and DCIR4(Clec4a1))

The human Dendritic cell immunoreceptor DCIR (hDICR) is expressed on myeloid cells, including classical and pDCs, monocytes, macrophages, granulocytes, and B lymphocytes (Flornes et al, 2004). The mouse DCIR family consist of four receptors: DCIR1, DCIR2, DCIR3 and DCIR4, all of which have different expression patterns (Flornes et al, 2004). mDCIR1 follows a similar expression pattern that hDCIR but it has not been found on T cells, and mDCIR2 appears to be a specific marker for CD8$^-$ DCs located in the red pulp and marginal zone of the mouse spleen (Kanazawa et al, 2002; Dudziak et al, 2007). mDCIR3 is present on lung alveolar macrophages and is co-expressed with mDCIR4 on tissue-resident macrophages from spleen, liver, peritoneum and small intestine (Hsu et al, 2017; Okada et al, 2020). mDCIR4 is also expressed on monocytes in different organs and the in vitro differentiation of Ly6C$^+$ cells from bone marrow into DCs and macrophages in the presence of GM-CSF and IL-4, reduced its expression (Hsu et al, 2017; Okada et al, 2020).

DCIRs mediate inhibitory signaling. hDCIR, mDCIR1 and mDCIR2 possess an ITIM motif in their intracellular domains that recruit both SHP-1 and SHP-2 phosphatases and is generally believed that this downregulates immune responses (Fig. 2) (Bates et al, 1999; Chen et al, 2024; Kanazawa et al, 2002; Richard et al, 2006). For example, targeting hDICR on pDCs or mDCIR1 on macrophages inhibits TLR9-induced pro-inflammatory cytokines (Meyer-Wentrup et al, 2008; Zhao et al, 2015). The CTLD of the receptor possesses an EPS motif, that recognizes mannotriose, sulfo-Lewis(a), Lewis(b) and Lewis(a) carbohydrates, and this lack of specificity is attributed to the recognition of the nonterminal N-glycan of these molecules (Lee et al, 2011; Bloem et al, 2014, 2013; Nagae et al, 2016).

DCIRs have been used to target antigens for T-cell presentation. For instance, antibody-mediated hDCIR targeting of pDCs triggers the internalization of the receptor in a clathrin-dependent manner and induces antigen presentation to T cells (Meyer-Wentrup et al, 2008; Klechevsky et al, 2010; Meyer-Wentrup et al, 2009). mDCIR2 targeting on DCs induce production of natural FoxP3$^+$ T-regulatory cells, acts as a regulatory receptor for the activation of CD8α$^-$ cDCs, and initiate extrafollicular B-cell responses to T-cell-dependent antigens (Chappell et al, 2012; Massoud et al, 2014; Uto et al, 2016).

DCIR has been implicated in protecting against the development of autoimmune diseases. For example, DCIR1 knockout mice spontaneously develop autoimmune diseases due to increased DC accumulation and T-cell activation, indicating a regulatory role for mDCIR1 (Fujikado et al, 2008). In addition, DCIR1 deficiency exacerbated two different models of RA, and the expression of this receptor is increased in the rheumatic joint of arthritic patients (Eklow et al, 2008). DCIR is also expressed on osteoclasts and binds to an asialo-biantennary N-glycan(s) on bone cells and myeloid cells, regulating autoimmune responses by downregulating M-CSF and RANKL signaling (Kaifu et al, 2021, 2023). In the experimental autoimmune encephalomyelitis model for multiple sclerosis, DCIR1 deficiency exacerbated the disease and the knockout mice presented with a higher number of infiltrated CD11c$^+$ DCs and CD4$^+$T cells in their spinal cords (Seno et al, 2015). Notably, there is an association between hDCIR polymorphisms and susceptibility to RA, systemic lupus erythematosus and primary Sjogren's syndrome (Guo et al, 2012; Lorentzen et al, 2007; Liu et al, 2015).

The role of DCIR has also been studied in chronic inflammatory conditions. Recently, it has been shown that DCIR1 is expressed on vascular residential macrophages that protect against atherosclerosis. Lack of the receptor on these cells causes dysfunctional cholesterol metabolism that exacerbates disease (Park et al, 2022). The functional role of DCIR has also been evaluated in models of ulcerative colitis, but the results are contradictory (Tokieda et al, 2015; Hütter et al, 2014). DCIR1 also protects against disease development in a mouse model of colorectal cancer and consistently, higher levels of hDCIR gene expression correlate with improved survival in colorectal cancer patients (Trimaglio et al, 2024).

The functions of DCIR appear to be detrimental during infections. For instance, *M. tuberculosis* infection is better controlled on DCIR1-deficient mice compared to wild-type animals (Troegeler et al, 2017). Mechanistically, through the JAK-STAT1 pathway, DCIR1 signaling sustained type I IFN responses in DCs, reducing Th1 differentiation during infection (Troegeler et al,

2017). In line with this, DCIR1 is also critical in the development of experimental cerebral malaria, since DCIR1-deficient mice present lower numbers of CD8 + T cells in the brain, less brain inflammation and were more protected from the disease (Maglinao et al, 2013). DCIR deficiency also protects against Theiler's murine encephalomyelitis virus, facilitating virus control in the brain and reducing neuropathology (Stoff et al, 2021). In addition, DCIR also recognizes HIV-1 gp140 glycoproteins, contributing to a productive virus infection of DCs, promoting virus propagation (Lambert et al, 2010; Bloem et al, 2014). Moreover, binding of HIV to human CD4$^+$ T cells isolated from patients upregulates DCIR expression (Lambert et al, 2010).

In summary, DCIR regulates immune cell activation through its inhibitory signaling pathway. Recognition of endogenous ligands by DCIR protects against the development of autoimmune diseases, but this receptor plays a deleterious role during infections.

### Dectin-2 (h:CLEC6A; m: Clec4n)

Dectin-2 is expressed on monocytes, macrophages, neutrophils and dendritic cells (Ariizumi et al, 2000; Taylor et al, 2005; Robinson et al, 2009). The cytoplasmic tail of this receptor interacts with the ITAM-bearing FcRγ chain to induce intracellular signaling, and this interaction is also required for Dectin-2 surface expression (Sato et al, 2006; Robinson et al, 2009). Following ligand recognition, Dectin-2 induces tyrosine phosphorylation of FcRγ, which recruits Syk and through PKCδ activates a signaling pathway similar to Dectin-1 (Fig. 2) (Gringhuis et al, 2011; Robinson et al, 2009; Sancho and Reis e Sousa, 2012b). Dectin-2 signaling also activates Casitas B-lineage lymphoma (c-Cbl), an E3 ubiquitin ligase. This molecule induces the ubiquitination and degradation of RelB, a noncanonical NF-κB subunit, allowing the translocation of the canonical NF-κB subunit p65 to the nucleus, downregulating immune responses (Zhu et al, 2016; Duan et al, 2021). Thus, c-Cbl acts as a negative regulator of Dectin-2 signaling pathways (Duan et al, 2021).

Dectin-2 contains a mannose-binding EPN motif and therefore, binds structures with high mannose content (Kerscher et al, 2013). Dectin-2 is able to recognize and participate in the host defense against various pathogens, such as serotype 3 *Streptococcus pneumoniae*, *Schistosoma mansonii*, *Pneumocystis* spp, *Histoplasma capsulatum*, *Paracoccidioides brasiliensis*, *Cryptococcus neoformans*, *Fonsecaea pedrosoi*, *Trichophyton rubrum*, *Microsporum audouinii*, *Malassezia furfur* and *Mucor* species, although the specific structures that are being recognized are not completely understood in all the cases (Akahori et al, 2016; Wüthrich et al, 2015; Kalantari et al, 2019; Haider et al, 2019; Thompson et al, 2021; Preite et al, 2018; Chang et al, 2017; Kottom et al, 2018; Kalantari et al, 2018; Tanno et al, 2019; Campuzano et al, 2020; Vendele et al, 2020; Ishikawa et al, 2013). Dectin-2 recognizes α-mannans present on the surface of *C. albicans*, and this interaction leads to the induction of pro-inflammatory cytokines, a NADPH oxidase-independent NET formation, and the development of adaptive immune responses (McGreal et al, 2006; Feinberg et al, 2013; Robinson et al, 2009; Saijo et al, 2010; Wu et al, 2019a). During C. *albicans* infection, mice lacking Dectin-2 have an increased susceptibility, with increased fungal burdens and rapid death compared with wild-type animals. Dectin-2 is also critical for the development of adaptive Th1 and Th17 protective responses to this fungus

(Robinson et al, 2009; Saijo et al, 2010). Interestingly, Dectin-2 triggers different responses between the hyphal and yeast form of *C. albicans*, probably through the recognition of different ligands expressed in the two morphological forms of the pathogen (Bi et al, 2010; Saijo et al, 2010; Sato et al, 2006). Recently, Dectin-2 has been shown to mediate trained immunity to C. *albicans* through the recognition of N-linked mannans (Rosati et al, 2024). Dectin-2 also plays a role in the immune response against A. *fumigatus* and a human deficiency in this receptor has been associated with invasive aspergillosis in an immunocompromised patient (Griffiths et al, 2021). In addition, Dectin-2 interacts with the O-linked α1,2 mannose chains located at the C-terminus domain of the glycoprotein Eng2, isolated from an attenuated vaccine strain of *Blastomyces dermatitidis*, and this interaction promotes adjuvant activity for vaccination (Wang et al, 2017; Wüthrich et al, 2021). Dectin-2-induced Th17 response is also important against M. *tuberculosis* infection in a process mediated by the recognition of the major lipoglycan, mannose-capped lipoarabinomannan (Man-LAM) (Yonekawa et al, 2014; Decout et al, 2018).

The discovery of the interaction between Dectin-2 and house dust mite (HDM) allergens from *Dermatophagoides farinae* and D. *pteronyssinus* lead to the study of the functional role of this receptor in allergic asthma (Barrett et al, 2009a). In fact, in response to HDM, Dectin-2 signaling produces cysteinyl leukotriene on DCs, initiates airway inflammation and induces Th2 and Th17 immune response in vivo (Barrett et al, 2009b, 2011; Clarke et al, 2014; Parsons et al, 2014; Norimoto et al, 2014). This immune response requires the PI3Kδ signaling pathway that leads to cysteine leukotriene production, and the secretion of the alarmin IL-33 (Lee et al, 2016b). Dectin-2 also recognizes LdpA, an A. *fumigatus* chitin-binding glycoprotein, inducing Th2-driven allergic airway inflammation (Muraosa et al, 2024).

Dectin-2 can also recognize endogenous molecules. It interacts with N-glycans of the β-glucuronidase present on dendritic cells, although the exact cellular localization of the glycoprotein recognized has not been clarified (Mori et al, 2017). In addition, Dectin-2 also can recognize Muc2, found in the small intestine and colon, but the consequence of this interaction has not been elucidated (Leclaire et al, 2018). Dectin-2 also mediates the phagocytosis of cancer cells by Kupffer cells, inhibiting liver metastasis, in a process dependent on the cell-surface transmembrane protein ERMAP and galectin-9 (Kimura et al, 2016; Chiffoleau, 2018; Li et al, 2023b).

Dectin-2 has been extensively studied in the context of infection, where it has been shown to play a protective role. Due to its interaction with molecules with a high mannose content, this receptor has the potential to be used as a drug target in the development of new therapies, particularly against fungal pathogens.

### MCL (Dectin-3; CLEC4D; CLECSF8)

Originally believed to be exclusively expressed on macrophages from mice and humans, MCL was subsequently shown to be expressed by neutrophils and monocytes from peripheral blood and weakly by several DC subsets (Balch et al, 1998; Graham et al, 2012). Although it is suggested that the receptor interacts with the FcRγ adaptor, the transmembrane domain of MCL does not contain a positively charged residue required for this association

and the mechanism of interaction is not completely understood (Miyake et al, 2013). Notably, MCL forms heterodimeric receptor with Mincle, and each receptor is required for the surface expression of its heterodimeric partner (Lobato-Pascual et al, 2013; Miyake et al, 2015; Zhu et al, 2013b; Kerscher et al, 2016b). MCL signals through Syk, inducing NF-κB activation through the CARD9–BCL100-MALT1 complex inducing phagocytosis, leading to the production of pro-inflammatory cytokines and ROS (Fig. 2) (Arce et al, 2004; Graham et al, 2012; Zhao et al, 2014).

MCL is required for the protection against *M. tuberculosis*. Infection in MCL-deficient mice results in higher bacterial burdens and increased mortality. There is also an association between a CLEC4D polymorphism identified in humans and an increased susceptibility to pulmonary tuberculosis (Wilson et al, 2015). Moreover, MCL is able to recognize trehalose-6,6'-dimycolate (TDM), a glycolipid present on the cell surface of mycobacteria that is also recognized by Mincle (Furukawa et al, 2013; Miyake et al, 2015). Mechanistically, TDM stimulation through Myd88 induces Mincle expression, which interacts with a constitutively expressed MCL to form the heterodimeric receptor that translocate to the cell surface (Kerscher et al, 2016a).

MCL interacts with fungi and initiates immune responses, recognizing α-mannans on the surface of *C. albicans*, *Paracoccidioides brasiliensis*, and *Cryptococcus* (Zhu et al, 2013b; Preite et al, 2018; Huang et al, 2018; Hole et al, 2016). MCL has been reported to form a heterodimeric receptor with Dectin-2, and the interaction with α-mannans on the surface of *C. albicans* hyphae is more effective in this state, than as their corresponding homodimers (Zhu et al, 2013). MCL fungal recognition is also involved in gut homeostasis. MCL-deficient mice are more susceptible to the DSS-induced model of colitis, and this is associated with higher fungal burdens of *Candida tropicalis* in the gut (Wang et al, 2016b). Importantly, the antifungal therapy of MCL-deficient mice was effective in treating colitis (Wang et al, 2016b). More recently, it was discovered that MCL deficiency increases tumorigenesis in a mice model of colorectal cancer, and that this was associated with an elevated *C. albicans* load (Zhu et al, 2021). MCL also participates in protection against bacterial pathogens, since it has been shown that MCL-deficient mice are more susceptible to *Klebsiella pneumoniae*, although the mechanism is not completely understood (Steichen et al, 2013).

MCL has also been studied in autoimmunity, playing a protective role in different diseases. For instance, specific MCL, Mincle or MCL/Mincle silencing in the central nervous system reduces clinical signs of EAE in rats (N'diaye et al, 2020). In addition, MCL deficiency limits pristane-induced lupus-like disease (Li et al, 2021). Mechanistically, lack of MCL promotes FoxO1-mediated apoptosis of myeloid-derived suppressor cells, reducing disease severity (Li et al, 2021).

The ability to form heterodimeric receptors with Dectin-2 and Mincle, makes MCL a versatile receptor involved in different aspects of immunity (see Box 1).

### Macrophage-inducible C-type lectin (Mincle; CLEC4E)

Originally discovered on macrophages based on its upregulation after pro-inflammatory stimulation, Mincle is expressed on monocytes, neutrophils, DCs and some subsets of B cells (Matsumoto et al, 1999; Kawata et al, 2012; Lee et al, 2012; Vijayan et al, 2012). Mincle forms a heterodimeric receptor with MCL, and

through a positively charged arginine residue in the transmembrane domain interacts with the adaptor FcRγ and signals through Syk, which activates the CARD9–Bcl-10– Malt1 and MAPK pathways leading to the expression of pro-inflammatory cytokines, chemokines and production of nitric oxide (Fig. 2) (Ishikawa et al, 2009; Kingeter and Lin, 2012; Lee et al, 2016c; Miyake et al, 2015; Schoenen et al, 2010; Strasser et al, 2012; Yamasaki et al, 2008a).

Mincle recognizes a broad variety of ligands, of both endogenous and exogenous origin, and it has been shown that the extracellular conformation of the receptor is key in the ability to bind different structures (Table 1). Mincle CTLD allows the binding of glucose residues in a $Ca^{+2}$-dependent manner, but in addition, presents a secondary binding site for carbohydrate binding, and a third hydrophobic region able to bind acyl chains (Feinberg et al, 2013; Jégouzo et al, 2014; Feinberg et al, 2016; Furukawa et al, 2013; Rambaruth et al, 2015).

Using biochemical fractionation it was demonstrated that Mincle recognizes the glycolipid TDM from *Mycobacterium* species, a molecule known for its adjuvant capacity, and this interaction required both the sugar and lipid recognition domains of the receptor (Ishikawa et al, 2009). Mincle also recognize the synthetic analog of TDM, Trehalose-6,6-dibehenate (TBM), and the presence of the receptor is required for the adjuvant ability of these molecules, that is lost in Mincle-deficient mice (Furukawa et al, 2013; Feinberg et al, 2013; Lu et al, 2018; Decout et al, 2017; Schoenen et al, 2010; Desel et al, 2013; Ostrop et al, 2015; Shenderov et al, 2013).

The role of Mincle in host immune response against pathogens depends on the microorganism being studied and, on the structures recognized. In vivo studies using *Mycobacterium bovis* BCG or *M. tuberculosis* Erdman, showed that the absence of Mincle lead to higher inflammation levels and increased mycobacterial loads (Behler et al, 2015; Lee et al, 2012). However, using *M. tuberculosis* H37Rv Mincle-deficient mice were able to induce a protective immune response and control the infection (Heitmann et al, 2013). Mincle is also required for mounting a protective immune response against *C. albicans, Pneumocystis, K. pneumoniae, Malassezia*, and *Tannerella forsythia*, where the receptor is involved in phagocytosis, inflammatory cytokine production and neutrophil extracellular trap formation (Chinthamani et al, 2017; Kottom et al, 2018; Sharma et al, 2014; Wells et al, 2008; Yamasaki et al, 2009). On the other hand, the interaction between Mincle and *Fonsecaea pedrosoi* and *F. monophora*, causative agents of chromoblastomycosis, is not sufficient to induce a protective immune response and contributes to chronicity of the infection, downregulating the immune response elicited by Dectin-1 and Dectin-2 against these pathogens (da Glória Sousa et al, 2011; Wevers et al, 2014; Wüthrich et al, 2015). In addition, Mincle has also a detrimental role during *Helicobacter pylori* and *Leishmania major* infection, where the receptor contributes to the pathogen survival limiting the induction of an effective immune response (Devi et al, 2015; Iborra et al, 2016). Mechanistically, it has been shown that after interaction with a proteinaceous ligand of *L. major*, Mincle recruits SHP-1 to the FcRγ chain reducing DCs activation and regulating immune responses (Iborra et al, 2016). Contrary to early findings suggesting that Mincle plays a protective role against pneumococcal pneumonia by recognizing the *Streptococcus pneumoniae* glycolipid glucosyl-diacylglycerol (Behler-Janbeck et al, 2016), more recent studies have shown that overexpression of Mincle can exacerbate

**Box 2 In need of answers**

1. How do CLRs within the Dectin-1 and Dectin-2 clusters interact and regulate signaling in response to ligands present on the same structure?
2. What are the mechanisms by which CLRs bind to different types of structures, and how do the recognized structures influence cellular responses?
3. What signaling pathways do CLEC-1 and LOX-1 use to mediate cellular responses?
4. Do heterodimeric receptors always cooperate when recognizing a shared ligand, and under what conditions might this cooperation vary?
5. How can the unique properties of these receptors be leveraged to accelerate the development of more effective therapeutic targets or vaccination strategies?

pneumococcal infection. This is due to the activation of the Nlrp3 inflammasome, leading to increased IL-1β secretion (Hollwedel et al, 2020).

Mincle is also involved in gut homeostasis, where interacts with mucosa-resident commensals present in Peyer patches, regulating the production of IL-17 by Th17 cells and group 3 innate lymphoid cells, and maintaining an optimal intestinal barrier function (Martínez-López et al, 2019). Moreover, Mincle-deficient mice present higher levels of gut bacterial translocation that leads to liver inflammation and deregulated lipid metabolism (Martínez-López et al, 2019). In vitro studies have shown that Mincle is able to interact with molecules derived from probiotic bacteria. For example, Mincle recognizes the cyclopropane-fatty acid α-glucosyl diglyceride from *Lactiplantibacillus plantarum*, and the Surface layer glycoproteins from *Lentilactobacillus kefiri* and *Levilactobacillus brevis*, suggesting that this receptor is involved in the modulation of the immune response elicited by these microorganisms (Malamud et al, 2019; Prado Acosta et al, 2021; Shah et al, 2016).

Mincle also regulates immune responses following recognition of endogenous ligands. For instance, Mincle has been reported to recognize Spliceosome-associated protein 130, a component of small nuclear ribonucloprotein, and β-glucosylceramide, an intracellular metabolite in the ceramide pathway (Yamasaki et al, 2008b; Nagata et al, 2017). These molecules are released after cell death and induce pro-inflammatory responses, leading to neutrophil accumulation in the damaged tissues (Nagata et al, 2017; Yamasaki et al, 2008a). Recently, it has been shown that β-glucosylceramide directly activates microglia through Mincle in Gaucher disease inducing phagocytosis of living neurons and exacerbating disease symptoms (Shimizu et al, 2023). β-glucosylceramide-Mincle interaction on T cells also increase Th17 cell proliferation and promotes EAE progression in mice (Zhang et al, 2022). In addition, human Mincle recognizes cholesterol crystals, structures present in the atherosclerotic plaques that induce inflammation and the activation of the NLRP3 inflammasome (Kiyotake et al, 2015; Lu et al, 2018). Mincle also interacts with cholesterol sulfate, a molecule present in the epithelial layer of barrier tissues, mediating inflammatory responses in a model of allergic dermatitis (Kostarnoy et al, 2017). There is also evidence suggesting that Mincle plays a critical role in regulating homeostasis, with SNPs in the receptor being implicated in several pathologies including Crohn's disease, multiple sclerosis, the non-alcoholic steatohepatitis (NASH)

chronic disease, RA, osteonecrosis, the pathogenesis of ischemic stroke and early brain injury after subarachnoid hemorrhage, obesity-induced adipose tissue inflammation and fibrosis and in pancreatic tumorigenesis (Tanaka et al, 2020, 2014; Suzuki et al, 2013; Ichioka et al, 2011; He et al, 2015; Andreev et al, 2020; Schierwagen et al, 2020; Gong et al, 2020b; N'diaye et al, 2020).

In summary, Mincle is capable of interacting with a wide range of molecules to induce different immune responses. Understanding the molecular mechanisms underlying the receptor–ligand interaction is important, if we are to utilize the functions of this receptor for therapeutic applications.

## Conclusion

C-type lectins are a diverse family of proteins involved in a wide range of functions in mammals. In this review, we have focused on the transmembrane CLRs of the Dectin-1 and Dectin-2 clusters, describing each receptor individually. These remarkable CLRs contain distinct intracellular signaling motifs and recognize a wide variety of endogenous and exogenous ligands, modulating multiple cellular responses such as endocytosis, cytokine and chemokine production, antigen presentation and cell migration. These responses in turn coordinate immunological processes that influence the outcome of infectious and non-infectious diseases, such as autoimmune and autoinflammatory conditions, and the maintenance of homeostasis. Understanding the molecular mechanisms involved in CLR ligand recognition and the signaling pathways triggered by these interactions will undoubtedly open promising opportunities for the development of improved therapeutic targets, new adjuvants and vaccination strategies (see Box 2).

## Peer review information

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

## Acknowledgements

The authors acknowledge funding from the Wellcome Trust (102705, 097377), Versus Arthritis (21164), Medical Research Council (MR/L020211/1), and the MRC Centre for Medical Mycology (MR/N006364/1).

## Author contributions

**Mariano Malamud**: Conceptualization; Writing—original draft; Writing—review and editing. **Gordon D Brown**: Conceptualization; Funding acquisition; Writing—original draft; Writing—review and editing.

## Disclosure and competing interests statement

The authors declare no competing interests.

