## [Peer Review File · EMBO Reports]

The Dectin-1 and Dectin-2 clusters: C-type lectin receptors with fundamental roles in immunity

Mariano Malamud and Gordon Brown

Corresponding author(s): Gordon Brown (gordon.brown@exeter.ac.uk), Mariano Malamud (m.g.malamud@exeter.ac.uk)

Review Timeline:

Submission Date:	22nd Jul 24
Editorial Decision:	16th Aug 24
Revision Received:	24th Sep 24
Accepted:	14th Oct 24

Editor: Achim Breiling

Transaction Report:

Dear Gordon,

Thank you for the submission of your review article to our editorial offices. I have now received the full set of referee reports that is copied below. As you will see, all three referees state that your manuscript is interesting and timely. However, they have several suggestions to improve the submission that I kindly ask you to address in a revised manuscript.

Given the constructive referee comments, I would thus like to invite you to revise your manuscript with the understanding that all referee points will be addressed in the revised manuscript and in a detailed point-by-point response.

I further have these editorial requests:

- Please add up to 5 keywords to the manuscript and place these below the abstract.
- We have space for 1 more figure, and it would be nice to have indeed 4 figures, as we encourage authors to maximize the use of visual elements, which will increase the accessibility of the piece to a non-specialist readership. Please consider adding 1 more figure and note the instructions regarding figures below. See also the suggestion by referee #1 regarding an additional figure.
- We usually ask our authors to include a box called "In need of answers" that briefly outlines the major questions that are still open in a given field in the form of a few bullet points. These questions can be accompanied by a brief explanation of what would be needed to address them and may provide helpful towards setting the stage for future experimentation in the field. For an example see this recent review we published: <https://www.embopress.org/doi/full/10.1038/s44319-024-00135-4>
- Please also add callouts for the box to the manuscript text (Box 1).
- Please also make sure the references and their callouts are formatted according to our reference format (with et al. for manuscripts with more than 10 authors):
<http://www.embopress.org/page/journal/14693178/authorguide#referencesformat>
- The manuscript, and in particular the reference list, are presently rather long. I wonder if this could be shortened a bit. But it is of course very important to cite primary research.

I think this is a very interesting review and while I appreciate that incorporating the referees' suggestions will still require some work, I am convinced that the article is worth it and will benefit from it.

When submitting your revised manuscript, we will require a Microsoft Word file (.doc) of the revised manuscript text including detailed figure legends (at the very end), but without the figures.

Please provide the final figures as separate, high-resolution files as .pdf, .eps, .tif, or .jpg (one file per figure). Please finalize the drafts provided and make sure they accurately illustrate the key scientific concepts that you wish to show.

Please also note the following points:

- If there are certain aspects of your figure draft that are based upon assumptions or where the scientific data remains ambiguous (for example, schematically depicting a presumed direct protein-protein interaction, protein shape or subcellular localizations etc.) please add a comment so that we can work with you on an accurate depiction. Please ensure the directionality and nature of interactions is presented accurately.
- If the figure or single panels of the figure have been adapted from a published figure, please add this information to the figure legend (e.g., 'Adapted from...' or 'Based on...'). The editor will discuss if a reference and permission will be necessary
- Please only re-use figures or parts of a figure if this is essential for understanding the concept communicated. Often a reference to a previous paper will suffice. If the figure contains re-used images or elements of images, including schematics, micrographs or photos, please make sure that you have the permission/license to publish it (this also applies to your own previous work, if the journal you published in retains copyright. Certain 'creative commons' open access licenses, such as CC-BY 4.0, allow re-use without additional formal permissions). All re-used material must be explicitly cited.
- If you use an image data base for scientific iconography (e.g., BioRender), please let us know if you have a license that allows for publication in an academic journal. Often authors use misleading iconography for expedience. Please ensure the information

shown is scientifically accurate. If in doubt, please discuss with the editor or provide a sketch so that our designers can create accurate iconography.

- For figures created using a software for editing vector objects like Inkscape, CorelDraw etc., please send the file as a PDF (or SVG, or EPS), PowerPoint or Keynote in which the labels and objects are still editable. For figures created using Adobe Illustrator, please send the Illustrator (.ai) file.

I look forward to seeing a revised version of your manuscript when it is ready. Please let me know if you have questions or comments regarding the revision.

Kind regards,

Achim

Referee #1:

In their review article entitled "The Dectin-1 and Dectin-2 clusters: C-type lectin receptors with fundamental roles in immunity", Malamud and Brown discuss each member of the Dectin clusters with regard to their cell subset-specific expression, ligands, signaling pathways and key functions. The review is really comprehensive, as it covers basically all relevant references and it also includes recent studies. The scope of the review article is of high relevance and the review convincingly highlights the wide range of functions that CLR's have in mammals. The focus on the Dectin-1 and Dectin-2 clusters makes sense, as both clusters are crucially involved in pathogen and/or self-antigen recognition, thus contribute to immune responses during infection and inflammatory processes. Overall, the review article is well written and an excellent contribution to the journal.

I suggest that the following minor points should be considered in a revised version:

1.) The part on MCL (Dectin-3) is rather short. It would make sense to extend this part by discussing the role of MCL in autoimmunity and inflammatory processes a bit more in detail.

2.) In the part on Mincle, some references are missing. For instance, the sentence "Mincle is also required for mounting a protective immune response against *C. albicans*, *Pneumocystis*, *K. pneumoniae*, *Malassezia*, and *Tannerella forsythia*" lacks the related references - please add. Moreover, the role of Mincle in bacterial recognition could be discussed a bit more in depth, for instance with regard to *S. pneumoniae* recognition.

3.) The summarizing table showing endogenous and exogenous ligands of CLR's of the Dectin-1 and Dectin-2 cluster is very helpful. However, I recommend highlighting the contents of this table also in an additional figure to facilitate readability. For example, selected ligands and/or pathogens and their interaction with the respective CLR's could be exemplified in a meaningful figure.

Referee #2:

This review covers aspects of ligand recognition by C-type lectin receptors, with a particular focus on each individual receptor that belongs to the clusters of Dectin-1 and Dectin-2. The discussion around each receptor is focused on the expression profile, described in literature ligands, in some cases co-receptors, ability to activate downstream signaling pathways and related cellular recognition. The review comprehensively covers the old and some recent literature on these receptors; however, it lacks an analysis of the mechanisms behind the activity of these receptors. In particular, when there is cross-talk between different receptors.

Overall, the classification of these receptors based on gene clusters or their structural homology is well-accepted in the field and it is often used in the reviews on this topic. Although this review provides some updated information and discusses some recently discovered receptors, in its structure and content, it does not stand out from other review (for example: Brown et al Nature Review 2018 or Reis e Sousa et al Immunity 2024). What if the authors do a more function-focused assessment of this family of receptors? It would be nice to see sections focused on the open questions, reasons and perspectives to investigate them, as well as novel approaches to access C-type receptors.

Specific comments:

- The organisation of Dectin-1 in nanoclusters has been reported (<https://pubs.acs.org/doi/10.1021/acs.jpcc.2c03557>). Could multimerization of Dectin-1 be the mechanism of the activation, without a need to form immune synapse?
- For an inhibitory receptor MICL, authors conclude that signaling pathways are not well investigated, but present an exciting opportunity for therapeutic intervention. Would it be possible to expand on which intervention is exactly envisioned? And which recent approaches can help to shed light on the downstream signaling of this receptor?
- Clec-2 can recognize O-glycans on podoplanin, and there is an interesting paper focused on the cross-talk of Dectin-1 and Clec-2, that authors can consider in their discussion (<https://elifesciences.org/articles/83037>)
- As mentioned above, could the author focus on the cross-talk between Dectin-1 and Dectin-2 clusters? It would be nice to review the literature aspects of the synergetic functions of these receptors. For instance, do Dectin-1 and Dectin-2 act synergistically to recognise the ligands on the same fugal particles? How does that regulate cell signaling?
- Figure 2 can be improved to be more aligned with the text in the article. The different classes of the receptors can be clustered and highlighted with color or boxes, or simply by making space between them. At the moment, the figure looks a bit crowded with information, and it is not easy to read the main message.
- Table 1 could have more condensed information and would benefit not only from ligand description. Authors can consider adding a column focused on the functional role of such interactions.

Referee #3:

This manuscript provides an extensive overview on the C-type lectins from the Dectin-1 and Dectin-2 gene clusters, covering the massive amount of literature on this topic.

Nevertheless, the manuscript comes across as slightly sloppy with some inconsistencies and grammatical errors. Also Figure 3 contains several mistakes (see below).

Specific comments:

- The CLEC-2 section is missing the interaction with Rhodocytin, which actually led to the discovery of CLEC-2 (Suzuki-Inoue Blood 2006). Rhodocytin is mentioned in the table, but deserves more attention in the main text, especially since new efforts are being undertaken to modify Rhodocytin for drug targeting (Obermann FASEB J 2024).
- Also worth mentioning here is the case of a CLEC-2 deficient individual with dysregulated lymphangiogenesis (Oishi Res Pract Thromb Haemost 2023).
- Heading for the LOX-1 section is missing.
- In the Dectin-2 cluster many differences exist between the mouse and human CLR. Therefore, I recommend to also add the mouse genes to the DCIR section, like is done for Dectin-2.
- I would opt to use MCL instead of Dectin-3, as twice as many papers refer to this CLR as MCL instead of Dectin-3 (same in the figures and Table 1). Also the CLECSF8 nomenclature for this receptor should be mentioned here.
- Figure 1: I found this figure confusing, as this figure suggests that all CLR are pro-inflammatory and thus boost immune responses. Actually, many are anti-inflammatory, yet this is not clear from the figure.
- Figure 2: I recommend to replace MelLec with CLEC-1, as this is the nomenclature used in the main text.
- Figure 3: This figure contains errors in the gene names for mouse Dectin-2 cluster. Clec4a1 is mentioned twice. The second Clec4a1 from the left should be replaced by Clec4a2. Clec6A is a human gene and should be replaced by Clec4n.
- Finally, a suggestion for the table. As some of the homologues are human or mouse-specific I would add here a column indicating expression of the CLR in human and/or mouse. DCIR now only mentions the human gene, but please add here also the 4 mouse genes.

Response to Reviewers comments:

We thank the Referees for the thoughtful and constructive evaluation of our manuscript. We have highlighted all major changes in yellow in the revised manuscript.

Referee #1:

In their review article entitled "The Dectin-1 and Dectin-2 clusters: C-type lectin receptors with fundamental roles in immunity", Malamud and Brown discuss each member of the Dectin clusters with regard to their cell subset-specific expression, ligands, signaling pathways and key functions. The review is really comprehensive, as it covers basically all relevant references and it also includes recent studies. The scope of the review article is of high relevance and the review convincingly highlights the wide range of functions that CLRs have in mammals. The focus on the Dectin-1 and Dectin-2 clusters makes sense, as both clusters are crucially involved in pathogen and/or self-antigen recognition, thus contribute to immune responses during infection and inflammatory processes. Overall, the review article is well written and an excellent contribution to the journal.

We thank the reviewer for these comments

I suggest that the following minor points should be considered in a revised version:

1.) The part on MCL (Dectin-3) is rather short. It would make sense to extend this part by discussing the role of MCL in autoimmunity and inflammatory processes a bit more in detail.

We have included this in a new paragraph.

2.) In the part on Mincle, some references are missing. For instance, the sentence "Mincle is also required for mounting a protective immune response against *C. albicans*, *Pneumocystis*, *K. pneumoniae*, *Malassezia*, and *Tannerella forsythia*" lacks the related references - please add. Moreover, the role of Mincle in bacterial recognition could be discussed a bit more in depth, for instance with regard to *S. pneumoniae* recognition.

We thank the reviewer for these suggestions which we have included in the new version of the manuscript.

3.) The summarizing table showing endogenous and exogenous ligands of CLRs of the Dectin-1 and Dectin-2 cluster is very helpful. However, I recommend highlighting the contents of this table also in an additional figure to facilitate readability. For example, selected ligands and/or pathogens and their interaction with the respective CLRs could be exemplified in a meaningful figure.

We thank the reviewer for this suggestion. We have included a new figure (Figure 4) with selected ligands recognized by the different receptors and the main responses induced by these interactions.

Referee #2:

This review covers aspects of ligand recognition by C-type lectin receptors, with a particular focus on each individual receptor that belongs to the clusters of Dectin-1 and Dectin-2. The

discussion around each receptor is focused on the expression profile, described in literature ligands, in some cases co-receptors, ability to activate downstream signaling pathways and related cellular recognition. The review comprehensively covers the old and some recent literature on these receptors; however, it lacks an analysis of the mechanisms behind the activity of these receptors. In particular, when there is cross-talk between different receptors.

Overall, the classification of these receptors based on gene clusters or their structural homology is well-accepted in the field and it is often used in the reviews on this topic. Although this review provides some updated information and discusses some recently discovered receptors, in its structure and content, it does not stand out from other review (for example: Brown et al Nature Review 2018 or Reis e Sousa et al Immunity 2024). What if the authors do a more function-focused assessment of this family of receptors? It would be nice to see sections focused on the open questions, reasons and perspectives to investigate them, as well as novel approaches to access C-type receptors.

We thank the reviewer for these comments. We have followed their suggestion and have included open questions that we believed will advance our understanding of the physiological role of the CLRs.

Specific comments:

- The organisation of Dectin-1 in nanoclusters has been reported (<https://pubs.acs.org/doi/10.1021/acs.jpcc.2c03557>). Could multimerization of Dectin-1 be the mechanism of the activation, without a need to form immune synapse?

We have included this a potential explanation of Dectin-1 signaling in its section.

- For an inhibitory receptor MICL, authors conclude that signaling pathways are not well investigated, but present an exciting opportunity for therapeutic intervention. Would it be possible to expand on which intervention is exactly envisioned? And which recent approaches can help to shed light on the downstream signaling of this receptor?

We have included the potential therapeutic opportunities that MICL offers as well as the techniques that could provide more information on the signalling pathway of this receptor.

- Clec-2 can recognize O-glycans on podoplanin, and there is an interesting paper focused on the cross-talk of Dectin-1 and Clec-2, that authors can consider in their discussion (<https://elifesciences.org/articles/83037>)

We have included a box called “Dectin-1 and Dectin-2 clusters crosstalk” in which we discuss this and other relevant papers on this important topic.

- As mentioned above, could the author focus on the cross-talk between Dectin-1 and Dectin-2 clusters? It would be nice to review the literature aspects of the synergetic functions of these receptors. For instance, do Dectin-1 and Dectin-2 act synergistically to recognise the ligands on the same fungal particles? How does that regulate cell signaling?

As mentioned above, we have added a box focused on the crosstalk between Dectin-1 and Dectin-2 clusters.

- Figure 2 can be improved to be more aligned with the text in the article. The different classes of the receptors can be clustered and highlighted with color or boxes, or simply by making space between them. At the moment, the figure looks a bit crowded with information, and it is not easy to read the main message.

We have followed the reviewer suggestion and color coded the CLRs based on their intracellular signaling

- Table 1 could have more condensed information and would benefit not only from ligand description. Authors can consider adding a column focused on the functional role of such interactions.

We thank the reviewer for this suggestion and now Table 1 contains a column focused on the immune effect of this interaction. In addition, we have now included a new figure with the immune effects of ligand-receptor interactions.

Referee #3:

This manuscript provides an extensive overview on the C-type lectins from the Dectin-1 and Dectin-2 gene clusters, covering the massive amount of literature on this topic.

Nevertheless, the manuscript comes across as slightly sloppy with some inconsistencies and grammatical errors. Also Figure 3 contains several mistakes (see below).

We thank the reviewer for these comments. We have corrected the manuscript in this new version.

Specific comments:

- The CLEC-2 section is missing the interaction with Rhodocytin, which actually led to the discovery of CLEC-2 (Suzuki-Inoue Blood 2006). Rhodocytin is mentioned in the table, but deserves more attention in the main text, especially since new efforts are being undertaken to modify Rhodocytin for drug targeting (Obermann FASEB J 2024).

We have included the interaction of Rhodocytin with CLEC-2 and the potential therapeutic use of modified rhodocytin.

- Also worth mentioning here is the case of a CLEC-2 deficient individual with dysregulated lymphangiogenesis (Oishi Res Pract Thromb Haemost 2023).

We have included this case in the appropriate section.

- Heading for the LOX-1 section is missing.
Apologise for this omission. The heading is now included.

- In the Dectin-2 cluster many differences exist between the mouse and human CLRs. Therefore, I recommend to also add the mouse genes to the DCIR section, like is done for Dectin-2.

We have included this suggestion in the relevant heading

- I would opt to use MCL instead of Dectin-3, as twice as many papers refer to this CLR as MCL instead of Dectin-3 (same in the figures and Table 1). Also the CLECSF8 nomenclature for this receptor should be mentioned here.

We have followed Reviewer's suggestion and modified Dectin-3 for MCL.

- Figure 1: I found this figure confusing, as this figure suggests that all CLR are pro-inflammatory and thus boost immune responses. Actually, many are anti-inflammatory, yet this is not clear from the figure.

We thank the reviewer for pointing this out. We have modified the figure to make it clear that CLRs modulate inflammation and not always boost immune responses.

- Figure 2: I recommend to replace MelLec with CLEC-1, as this is the nomenclature used in the main text.

We have modified the nomenclature in the figure for consistency.

- Figure 3: This figure contains errors in the gene names for mouse Dectin-2 cluster. Clec4a1 is mentioned twice. The second Clec4a1 from the left should be replaced by Clec4a2. Clec6A is a human gene and should be replaced by Clec4n.

We apologise for the mistakes in this figure. We have corrected them in the new version.

- Finally, a suggestion for the table. As some of the homologues are human or mouse-specific I would add here a column indicating expression of the CLR in human and/or mouse. DCIR now only mentions the human gene, but please add here also the 4 mouse genes.

We have followed reviewer's suggestions and now the table includes expression of both human and/or mouse genes, as well as the immune function of ligand-receptor interactions.

Prof. Gordon Brown
University of Exeter
MRC Centre for Medical Mycology
Geoffrey Pope Building
Stocker Road
Exeter EX4 4QD
United Kingdom

Dear Prof. Brown,

I am pleased to inform you that your manuscript has been accepted for publication in EMBO reports. Your manuscript will be processed for publication by EMBO Press. It will be copy edited and you will receive page proofs prior to publication.

You will soon be contacted by Springer Nature to sign your publishing license. When you login to the customer service website, please use the token/code copied below to waive the article publication charges. Should you experience any difficulty, please email publishing@embo.org.

Token/code: LTEXNDYZODG3MTC

If you have any other questions, please do not hesitate to contact the Editorial Office. Thank you for your contribution to EMBO Reports.

Yours sincerely,
